# Theta rhythmicity governs human behavior and hippocampal signals during memory-dependent tasks

Marije ter Wal [1✉], Juan Linde-Domingo [1,2], Julia Lifanov[1], Frédéric Roux[1], Luca D. Kolibius[1,3], Stephanie Gollwitzer[4], Johannes Lang [4], Hajo Hamer [4], David Rollings[5], Vijay Sawlani[5], Ramesh Chelvarajah[5], Bernhard Staresina [1,6], Simon Hanslmayr [1,3] & Maria Wimber [1,3✉]

Memory formation and reinstatement are thought to lock to the hippocampal theta rhythm, predicting that encoding and retrieval processes appear rhythmic themselves. Here, we show that rhythmicity can be observed in behavioral responses from memory tasks, where participants indicate, using button presses, the timing of encoding and recall of cue-object associative memories. We find no evidence for rhythmicity in button presses for visual tasks using the same stimuli, or for questions about already retrieved objects. The oscillations for correctly remembered trials center in the slow theta frequency range (1-5 Hz). Using intracranial EEG recordings, we show that the memory task induces temporally extended phase consistency in hippocampal local field potentials at slow theta frequencies, but significantly more for remembered than forgotten trials, providing a potential mechanistic underpinning for the theta oscillations found in behavioral responses.

[1] School of Psychology & Centre for Human Brain Health, University of Birmingham, Edgbaston, B15 2TT Birmingham, UK. [2] Max Planck Institute for Human Development, 14195 Berlin, Germany. [3] Centre for Cognitive Neuroimaging, School of Psychology and Neuroscience, University of Glasgow, G12 8QB Glasgow, UK. [4] Universitätsklinikum Erlangen, 91054 Erlangen, Germany. [5] Complex Epilepsy and Surgery Service, Queen Elizabeth Hospital Birmingham, Edgbaston, B15 2GW Birmingham, UK. [6] Department of Experimental Psychology, University of Oxford, OX2 6GG Oxford, UK. ✉email: m.j.terwal@bham.ac.uk; maria.wimber@glasgow.ac.uk

In everyday life, our brains receive a virtually never-ending stream of information that needs to be stored for future reference or requires integrating with pre-existing knowledge. The hippocampus is the hub where encoding and retrieval of information is coordinated (for reviews see refs. [1–3]). Information streams within hippocampus and between hippocampus and cortex are thought to be orchestrated by the phase of the theta rhythm[4–6]. Here, we ask whether theta oscillations clock responses during memory tasks, producing rhythmicity in behavior.

During memory formation, information processed by cortical regions is sent to the hippocampus and presumably encoded in the form of a sparse, associative index. Conversely, during retrieval, cues trigger the completion of existing patterns encoded in hippocampus, eliciting reinstatement of the memory in associated cortical regions. Both memory encoding and retrieval have been associated with changes in oscillatory patterns in hippocampal local field potentials (LFPs). The LFP of rodents is dominated by oscillations in the 4–8 Hz theta frequency band, while a broader low-frequency band is apparent in humans, with frequencies in intracranial recordings often peaking between 1 and 5 Hz during memory tasks[7–10]. Several studies have shown that encoding of later-remembered items is accompanied by higher theta power compared to later-forgotten items[9,11–13], but see ref. [14]. Similarly, phase–amplitude coupling between theta and gamma oscillations increases during successful encoding[12,15,16]. Finally, spiking activity of hippocampal neurons was reported to lock to the LFP at theta frequencies[17] specifically during successful encoding[18].

During memory retrieval, theta power increases in cortical areas that are involved in reinstatement[19] and synchronization between these areas and hippocampus increased at theta frequencies[20–24]. Intriguingly, recall signals in hippocampus precede reinstatement in the cortex by about one theta cycle, suggesting hippocampus and cortex communicate within theta "windows" during memory recall[3]. In recent human studies, reinstatement of remembered associations was found to be theta-rhythmic[25], and remembered spatial goal locations were represented at different phases of the theta rhythm[26,27].

Theoretical work in the memory domain proposes that destructive interference between new information entering hippocampus and stored, reactivated information is reduced by locking to opposing theta phases[28]. Indeed, strengthening of synaptic connections (long-term potentiation) is more likely to occur around the trough of theta cycles[29,30], while synaptic depression is more pronounced at the peak[30]. In line with these findings, rodent work suggests that communication of new information from cortex to hippocampus predominantly occurs around the theta trough, while retrieval-related spiking activity in hippocampus is observed around the theta peak[31–34]. Intracranial recordings from epilepsy patients suggest similar network dynamics, with entorhinal cortex and hippocampus synchronizing their theta phase during encoding, while hippocampus locked to the downstream subiculum during retrieval[35]. Furthermore, optogenetically suppressing neural activity during task-irrelevant phases of the theta oscillation improves performance[36], demonstrating that the theta phase has a functional link to memory performance.

Consistent locking of encoding and retrieval processes to the theta rhythm predicts that these processes appear as rhythmic. Rhythmicity might therefore be visible in behavioral markers that depend on long-term memory. To our knowledge, no work has tested for such rhythmicity in memory-dependent tasks. However, recent studies on attentional scanning in both monkeys and humans suggest that oscillatory activity can manifest in behavioral performance, reflecting periodic switches in attended locations[37–42].

Here, we ask whether the presumed clocking of neural memory processes by the theta rhythm translates into an observable oscillatory modulation of behavior. We analyze responses from hundreds of participants completing a memory task, in which they press buttons to indicate the exact time points at which they formed or recalled associative memories. We find significant oscillations in both encoding and retrieval responses, with peak frequencies in the lower theta frequency band (1–5 Hz[8,9]). No oscillatory signatures are observed in button presses from task phases that do not depend on memory. Moreover, incorrect trials do not lock to the rhythm identified for correct trials. To underpin our behavioral findings with a neural mechanism, we analyze hippocampal LFPs recorded in epilepsy patients. These exhibit temporally extended phase locking in the low theta range during memory-dependent task phases, for correct but not incorrect trials. Finally, we show that encoding and retrieval trials show maximal phase alignment at opposite phases of the theta rhythm. Together, our results demonstrate that theta-rhythmicity of memory processing can be detected in human behavior and direct hippocampal recordings.

## Results

**Button presses indicate the timing of memory-dependent and -independent processing.** In this study we asked whether signatures of hippocampal rhythms can be found in behavioral responses during memory encoding and retrieval. We analyzed the data from 226 participants who performed associative memory tasks, consisting of multiple blocks with encoding, distractor, and retrieval phases (Fig. 1A). During encoding phases, participants viewed a cue (verb or scene image), followed by a stimulus (photo or drawing of an object). They pressed a button when they made an association between cue and stimulus, providing us with an estimate of the timing of memory formation (Encoding button press). During retrieval phases, cues were shown in random order and participants were asked to remember the associated objects. Participants in group 1 ($n = 71$) indicated the moment they remembered the object by pressing a button (Retrieval button press) and then answered one or two catch questions (e.g., "animate or inanimate?") about the already reinstated object (Catch-after-retrieval button press). Participants in group 2 ($n = 155$) were shown the catch question before the cue appeared. This group mentally reinstated the object and pressed the button as soon as they were able to answer the question (Catch-with-retrieval button press), indicating the time of subjective memory retrieval in this group. Each participant memorized between 64 and 128 cue–object pairs. Objects, cues, and catch questions varied between experiments; for details see "Methods" and Supplementary Table 1.

In order to separate memory processes from perceptual task elements, a separate group (group 3; $n = 95$) performed visual control tasks using the same questions and objects (Fig. 1B). Participants were shown a question (e.g., "animate or inanimate") followed by an object, and they answered the question by pressing a button (Visual button press). Note that the button presses from the visual task do not depend on episodic memory, as they pertain to objects that are constantly visible. Answers to the catch question for memory group 1 (Catch-after-retrieval button press) are also not expected to rely on hippocampal memory retrieval, since they are asked after objects are reinstated. Answering these questions is, however, likely to rely on maintenance of the retrieved object in working memory. We use the term memory-dependent as relying on hippocampus-dependent associative memory.

We analyzed performance of each participant based on the catch questions. Participants who performed at chance level

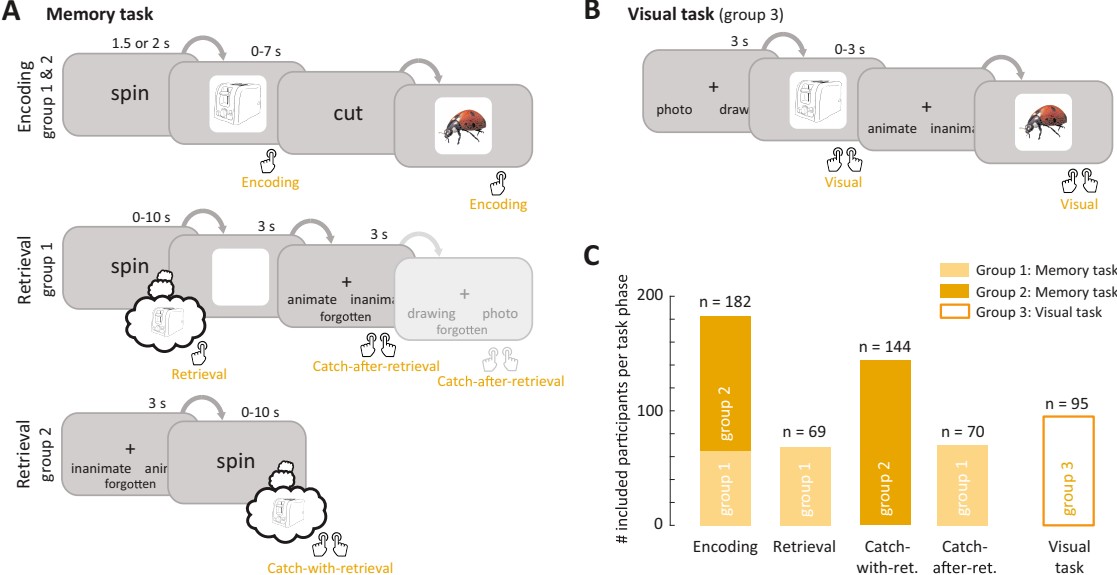

**Fig. 1 Button presses indicate the timing of memory-dependent and -independent processing. A** Structure of the memory task. Two groups of participants (groups 1 and 2; $n = 226$) completed blocks consisting of an encoding phase (top row) in which they associated cues ("spin", "cut") to objects, a distractor phase (not shown), and one of two versions of a retrieval phase (bottom rows), in which they answered catch questions about remembered objects ("animate or inanimate?", "photo or drawing?"). **B** Structure of the visual task. A separate group of participants (group 3; $n = 95$) answered questions about objects on the screen, using the same questions and stimuli as the memory task. **C** Number of participants that were included in further analyses, after exclusion of participants with a high number of incorrect and/or timed-out trials (Supplementary Fig. 1A). Note that participants in group 1 contributed button presses to three task phases (Encoding, Retrieval, and Catch-after-retrieval), group 2 contributed to 2 (Encoding and Catch-with-retrieval), and group 3 to 1 task phase (Visual). The example stimuli shown in **A** and **B** were taken from the BOSS database[70].

(binomial test) were excluded from further analyses ($n = 12$ for memory task; $n = 0$ for visual task, Supplementary Fig. 1A). In addition, 32 (1) participants with sufficient performance had a low trial count for the encoding (retrieval) phase due to trial time-outs, and hence were excluded for encoding (retrieval) phase analyses (Supplementary Fig. 1A). In general, participants responded well within the allotted response times (Supplementary Fig. 1C and Supplementary Table 3). The included participants (Fig. 1C) had an average performance of 84.0% (range 56.3–100%) for the memory groups and 96.3% (range 78.2–100%) for the visual group (Supplementary Fig. 1B). For the number of responses and reaction times per task phase see Supplementary Fig. 1 and Supplementary Table 3.

**Oscillatory patterns can be detected in behavioral responses using the O-score.** The button presses from the memory tasks provided us with estimates of when participants formed and reinstated memories on each trial. Figure 2A shows the button presses from all retrieval trials of one participant, as well as the smoothed response density across trials. We asked whether the response densities showed oscillations, as suggested by the trend-removed trace in Fig. 2A (right), and whether these patterns differed between memory-dependent and -independent task phases.

To address this we used the Oscillation score[43]. This procedure identifies the dominant frequency in the response time stamps, and provides a normalized amplitude at this frequency: the O-score. In brief, after removal of early and late outliers (Fig. 2B, step I), we computed the O-score following the procedure in ref. [43] (Fig. 2B, blue box): The auto-correlation histogram (ACH) is computed for the button presses from correct trials and smoothed with a Gaussian kernel ($\sigma = 2$ ms) to reduce noise. The central peak of the ACH is removed. All remaining positive lags are Fourier transformed, and the frequency with the highest magnitude is found within a frequency range of interest (adjusted

per participant based on the signal length (lower bound) and number of responses (upper bound), with a minimum of 0.5 Hz and maximum of 40 Hz). The O-score is computed by dividing the peak magnitude by the average of the entire spectrum.

The O-score indicates how much the spectral peak stands out, but does not take into account the overall response structure (gray trend curves in Fig. 2) and the limited number of data points, which could introduce a frequency bias. To account for this, we fitted a trend curve (Gamma distribution) for each participant and generated 500 random series of button presses based on this structure, with the same number of data points as the original dataset (Fig. 2B, II, see "Methods" for details). We computed the O-score at the peak frequency from the intact data for each randomization, and Z-transformed the original O-score against the 500 reference O-scores (Fig. 2B, III). This allowed us to statistically assess the strength of the behavioral oscillation for each participant and task phase, and perform second-level statistical assessments across participants.

We validated the performance of the O-score and Z-scoring methods using simulated data that mimicked the characteristics of the behavioral dataset (Supplementary Note 2 and Supplementary Fig. 14). This provided several important validations: (1) when no or very weak oscillations were present in the simulated data, the O-score was never significant at the population level; (2) when the O-score reached significance for our simulated populations, the O-score identified the correct frequency; and (3) the O-score procedure performed well for different task phases, despite differences in participant count, number of responses, or average reaction time.

**Behavioral responses oscillate at theta frequencies for memory-dependent task phases.** Significant O-scores were observed for encoding and retrieval button presses from both versions of the memory task (Fig. 3A), specifically Encoding ($t(181) = 6.20$, $p < 0.001$); Retrieval ($t(68) = 4.58$, $p < 0.001$); and Catch-with-

## A  Example participant

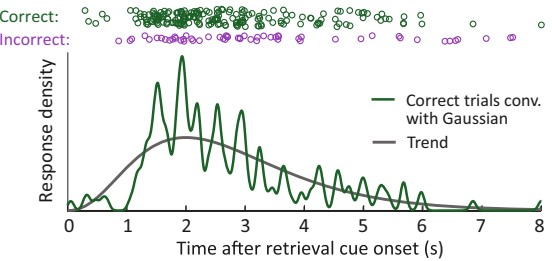

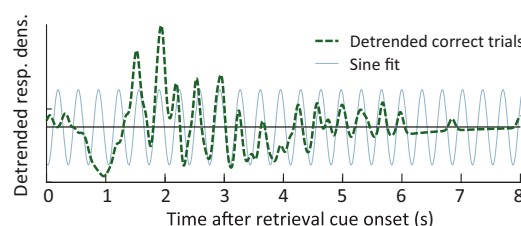

## B  Oscillation score for example participant from panel A

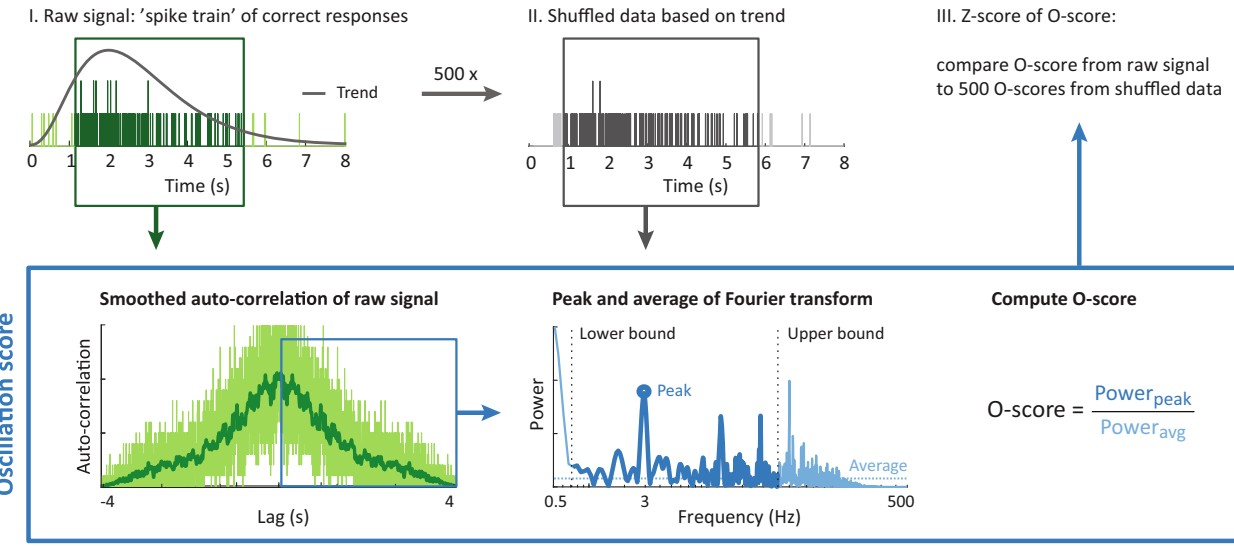

**Fig. 2 Oscillatory patterns can be detected in behavioral responses using the *O*-score. A** Timing of retrieval button presses for an example participant. Each circle is one button press, with correct trials in green and incorrect trials in purple. Convolving the correct trials with a Gaussian kernel (left panel, solid green line) reveals an overall trend in response density (left panel, gray line) as well as an oscillatory modulation (right panel, dashed green line). This oscillation was well fitted by a sine wave at the frequency identified by the Oscillation score procedure (light blue, arbitrary amplitude). More examples are given in Supplementary Fig. 2. **B** Step-by-step representation of the Oscillation score method, for the participant from panel **A**. *O* scores were computed for the original data (dark green) and 500 reference datasets with the same overall response trend (gray) following the procedure from ref. [43], summarized in the blue box. For parameter validation see Supplementary Fig. 3, and for further details see the main text and "Methods". Source data are provided as a Source Data file.

retrieval ($t(143) = 5.08$, $p < 0.001$; all Bonferroni-corrected for five comparisons; effect sizes in Supplementary Table 5). Additionally, the proportion of participants with significant *O*-scores was high (Fig. 3B): 74.7% for Encoding, 69.6% for Retrieval, and 76.4% for Catch-with-retrieval. On the other hand, no evidence for a behavioral oscillation was found for memory-independent task phases, with non-significant *O*-scores for the catch questions after reinstatement and the visual task (Catch-after-retrieval: $t(69) = 1.69$, $p = 0.240$; Visual: $t(94) = -4.10$, $p = 1.00$; Bonferroni-corrected for five comparisons; the *t*-value captures deviation from the reference-defined threshold; hence, both non-significant and negative *t*-values signify lack of evidence for oscillations). Note that the Catch-after-retrieval data were obtained from the participants in memory task group 1, while the Visual task was recorded in an independent group of participants (see Fig. 1C). The proportion of participants with significant *O*-scores was lower than for memory-dependent phases: 64.3% for Catch-after-retrieval and 37.9% for the Visual task. Raw *O*-scores showed a similar pattern across task phases (Supplementary Fig. 4A).

To test whether memory-dependent task phases had significantly higher *O*-scores than memory-independent phases, we fitted a linear mixed-effects model to the *Z*-scored *O*-scores. Fixed terms in this model were memory dependence and length of the time series, which varied substantially between task phases (Supplementary Fig. 1C); we included the intercept per subject as random effect, to address potential dependencies due to participants of the memory task contributing 2 or 3 data points. We found strong support for an effect of memory dependency on *O*-score, with significantly higher *Z*-scores for memory-dependent than memory-independent task phases (Fig. 3A; coefficient = 0.28; 95% CI: 0.19–0.36; $t(556) = 6.55$; $p < 0.001$). This was unaffected by time series length (coefficient = 0.0022; 95% CI: −0.0035 to 0.0080; $t(556) = 0.768$; $p = 0.443$). Post hoc (paired) *t*-tests confirmed these trends within memory groups 1 and 2, and demonstrated that the visual task had significantly lower *O*-scores than all other task phases (Supplementary Fig. 4B–D and Supplementary Table 4, effect sizes in Supplementary Table 5). These trends were qualitatively similar within all included experiments (see Fig. 4E) and were hence not driven by a single experimental setup or stimulus set.

The high *O*-scores we found for memory-dependent task phases are a strong indication of rhythmicity of behavior. Interestingly, the peak frequencies of significant *O*-scores from

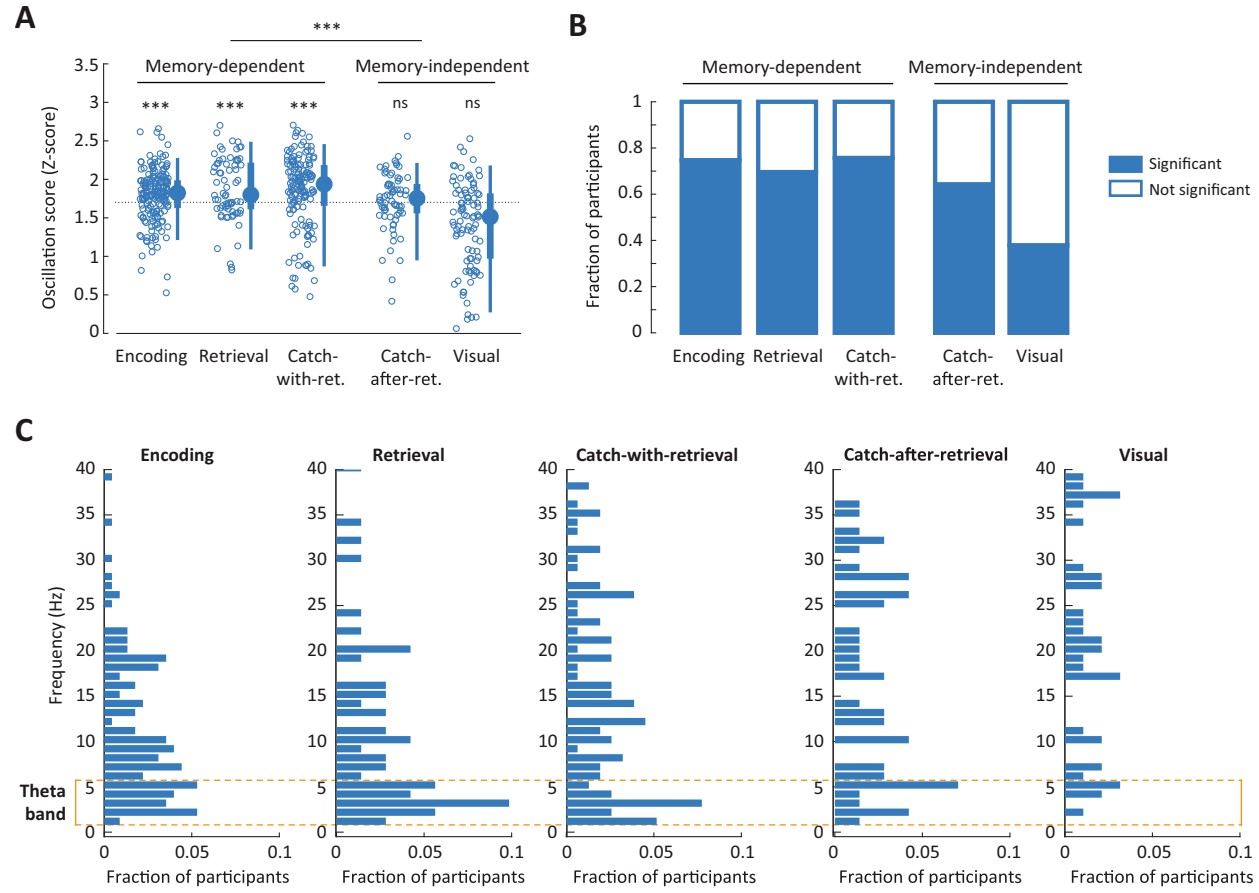

**Fig. 3 Behavioral responses oscillate at theta frequencies for memory-dependent task phases. A** Scatter plot of $O$-scores ($Z$-scored) per task phase, where each circle is one participant, and box plots representing the 5, 25, 50, 75, and 95% bounds of the $O$-score distribution across participants. The dashed line gives the significance threshold for single participants ($\alpha = 0.05$, one-tailed, $Z$-distribution). The outcome of a second-level $t$-test is given above each task phase, and the comparison between memory-dependent and -independent task phases is given at the top (linear mixed model, see the main text and "Methods"). See Supplementary Fig. 4 for raw $O$-scores and additional statistics and Supplementary Fig. 5 for analyses excluding participants for whom the response density trend could not be fitted; n.s. not significant; *: $0.05 \geq p > 0.01$; **: $0.01 \geq p > 0.001$; ***: $p \leq 0.001$ (for exact values see the main text), Bonferroni-corrected for five comparisons. **B** Proportion of participants with significant (i.e. above the $Z$-threshold in **A**; blue bars) and non-significant $O$-score (white bars). **C** Histograms of peak frequencies per task phase, for participants with significant $O$-scores, as a fraction of the total number of participants. The yellow outline indicates the 1–5 Hz frequency band. Participant numbers can be found in Fig. 1C. Source data are provided as a Source Data file.

memory-dependent task phases were non-uniformly distributed (Kolmogorov–Smirnoff test for uniformity; Encoding: $D^* = 0.395$, $p < 0.001$; Retrieval: $D^* = 0.381$, $p < 0.001$; Catch-with-Retrieval: $D^* = 0.236$, $p < 0.001$; corrected for five comparisons). Most participants showed peak frequencies (Fig. 3C) between 1 and 5 Hz or harmonics of this range. These frequencies align with the low theta band identified in human hippocampal recordings during memory tasks[8,9]. Conversely, peak frequencies were broadly distributed for Catch-after-retrieval and the Visual task (Kolmogorov–Smirnoff test for uniformity; Catch-after-Retrieval: $D^* = 0.158$, $p = 0.957$; Visual: $D^* = 0.0894$, $p = 1.00$; corrected for five comparisons). To directly test for a difference between the frequencies of memory-dependent and -independent task phases, we fitted a linear mixed model to the frequencies of significant $O$-scores, with memory dependency and time series length as fixed effects, and participant as random effect. This revealed that frequencies for memory-dependent task phases were significantly lower than for memory-independent tasks (coefficient = $-5.09$; 95% CI: $-7.52$ to 2.67; $t(371) = 6.55$; $p < 0.001$, post hoc tests in Supplementary Table 6). There was a small effect of time series length on frequency (coefficient = $-0.202$; 95% CI: $-0.358$ to 0.0461; $t(371) = -2.55$; $p = 0.011$), with higher

frequencies for memory-independence corresponding to shorter time series. To ensure that the results were not amplified by the lower-frequency limit in the $O$-score procedure, set to 1/3 of the time series length, we loosened this bound to twice the time series length and recomputed the $O$-scores. This produced similar results (Supplementary Fig. 6; Catch-after-retrieval: $t(69) = 0.908$, $p = 0.917$; Visual: $t(94) = -6.20$, $p = 1.00$; Bonferroni-corrected for five comparisons), reaffirming that the identified difference between memory-dependent and -independent task phases is not caused by differences in response times.

**Reaction times of incorrect trials are not locked to the behavioral oscillation.** The $O$-scores reported in Fig. 3 were based on correct trials only. Due to a low number of incorrect trials, it was not possible to establish whether incorrect trials show oscillatory modulation. However, we were able to test whether incorrect trials locked to the oscillation of the correct trials (correcting for fitting bias, see below) for every participant with a significant $O$-score. The instantaneous phase of the oscillation was determined by smoothing and filtering the correct response trace around the participant's peak frequency (example in Fig. 2A, solid green line) and performing a Hilbert transform. We then determined the

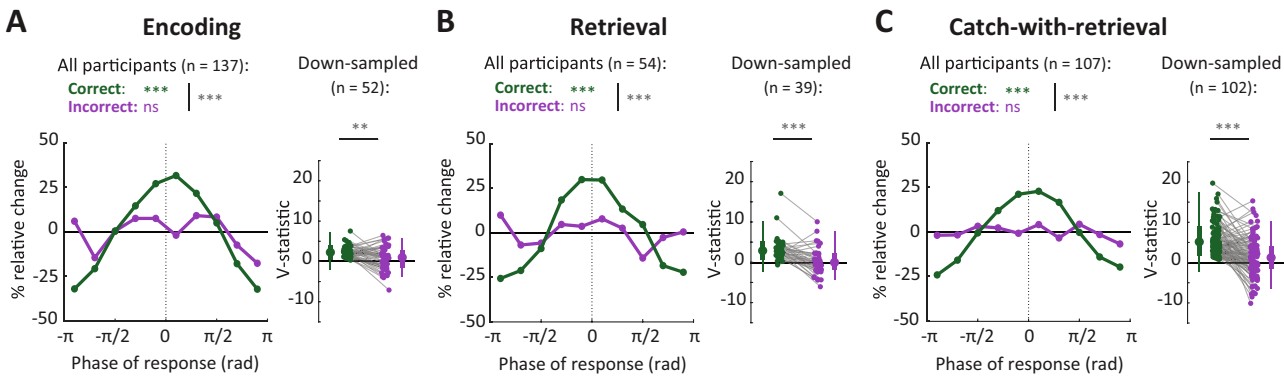

**Fig. 4 Reaction times of incorrect trials are not locked to the behavioral oscillation.** Phase distributions of incorrect responses relative to all correct responses (purple) and of correct responses relative to all other correct responses (green) for the task phases with significant *O*-scores: Encoding (**A**), Retrieval (**B**), and Catch-with-retrieval (**C**). The left panels shown deviations from uniform phase distributions across all participants with significant *O*-scores. Here, 0 radians is defined as the peak of the oscillation, and ±π as the trough. Statistics for correct and incorrect trials individually were obtained with a *V*-test for non-uniformity of the distribution around phase 0, and a permutation test was used to compare correct with incorrect distributions to a trial label-shuffled reference distribution (500 permutations, see "Methods"). Right panels show *V*-statistics for correct and incorrect trials of participants with at least 10 incorrect trials (each gray line is one participant), after down-sampling the number of correct trials to the number of incorrect trials. Shown are the mean *V*-statistics per participant (dots) and the distribution of *V*-statistics across all participants (box plots indicating the 5, 25, 50, 75, and 95% bounds of the distributions), which were compared with a two-tailed paired *t*-test. Phase distributions corresponding to these down-sampled datasets can be found in Supplementary Fig. 7. n.s. not significant; *: $0.05 \geq p > 0.01$; **: $0.01 \geq p > 0.001$; ***: $p \leq 0.001$ (for exact values see the main text), Bonferroni-corrected for three comparisons. Source data are provided as a Source Data file.

phases at which the incorrect button presses occurred (Fig. 4, purple lines). Similarly, we found the phase of each correct response relative to all other correct trials, by recomputing the instantaneous phase without the trial of interest, avoiding circularity (Fig. 4, green). As expected, for all memory-dependent task phases, correct trials more often occurred around the peak of the oscillation (*V*-test for non-uniformity around 0°; Encoding: $V$ (11398) = 1861.1, $p < 0.001$; Retrieval: $V$ (8036) = 1174.3, $p < 0.001$; Catch-with-retrieval: $V$ (23815) = 2847.5, $p < 0.001$; Bonferroni-corrected for three comparisons; Note that the high trial count can inflate test results). On the other hand, phase distributions were uniform for incorrect trials (Encoding: $V$ (1270) = 44.1, $p = 0.120$; Retrieval: $V$ (954) = 11.2, $p = 0.911$; Catch-with-retrieval: $V$ (5328) = 73.8, $p = 0.229$; Bonferroni-corrected for three comparisons). This suggests that incorrect responses did not lock to the rhythm of the correct trials, while correct responses were locked to the oscillation from other correct trials, pointing to the behavioral relevance of the identified oscillation. We did not perform this analysis for memory-independent task phases, as we found no evidence for oscillations.

We next directly tested the phase modulation of correct versus incorrect trials, accounting for potential biases caused by differences in trial count and procedure. We shuffled correct and incorrect trial labels 500 times per participant (i.e. keeping the original trial counts and response times), and computed the *V*-statistics of shuffled-correct and shuffled-incorrect trials as described previously. The difference in phase modulation between real-correct and real-incorrect responses was significantly higher than expected based on the difference between the shuffled-correct and -incorrect trials (Encoding: $p < 0.002$; Retrieval: $p < 0.002$; Catch-with-retrieval: $p < 0.002$).

For participants with at least 10 incorrect trials we also compensated for trial number biases by subsampling the number of correct trials to the number of incorrect trials (repeated 100 times), and recomputing the phases of both the selected correct and the incorrect trials relative to the remaining correct trials (Fig. 4, right panels). This procedure also demonstrated significantly higher phase modulation for correct than for incorrect trials for each of the memory-dependent task phases

(two-tailed paired *t*-test; Encoding: $t(51) = 4.07$, $p < 0.001$, 95% CI: <0.0001–0.14; Retrieval: $t(38) = 5.41$, $p < 0.001$, 95% CI: <0.0001–0.024; Catch-with-retrieval: $t(101) = 7.25$; $p < 0.001$; 95% CI: <0.0001; Bonferroni-corrected for three comparisons). In conclusion, all comparisons show that correct responses are substantially more phase-locked to each other than to incorrect trials. Note that we cannot rule out that incorrect trials lock to each other at a different frequency. Combining these findings with our previous analyses, our data suggest that correct trials show substantial behavioral oscillations, but that incorrect trials do not lock to this oscillation.

**Increased phase locking of hippocampal LFPs during encoding and retrieval.** The data reported so far indicate that across trials, memory-relevant behavioral responses fall onto a consistent phase of a theta oscillation. The presence of such an oscillation, determined on the basis of one response per trial, implies phase consistency across trials in the neural oscillations in hippocampus presumed to underly memory formation and reinstatement, as previously shown by Kota et al.[44], and Fell et al.[45]. We hypothesized that this phase consistency, induced by events in the trial, persists until the participant successfully encodes or retrieves the memory (expected to slightly precede the button press).

To test these predictions, we recorded hippocampal LFPs in 10 epilepsy patients undergoing seizure monitoring using intracranial EEG. These patients performed the same memory task as healthy participants, and their behavioral data are included in the previous results. We recorded from 42 Behnke–Fried microelectrodes located in hippocampus (Fig. 5A), which ensures a truly local hippocampal signal, minimizing influence of volume conduction from neighboring cortical regions and connections. We wavelet-transformed the LFPs and computed the pairwise phase consistency (PPC; ref. [46]) across trials for every frequency and time point. The PPC quantifies how similar the LFP phases are across trials. We performed this analysis separately for cue-, stimulus- and response-locked data and for correct (Fig. 5B) and incorrect trials (Fig. 5C).

In line with our predictions, PPC across correct trials significantly increased after stimulus onset for encoding, and

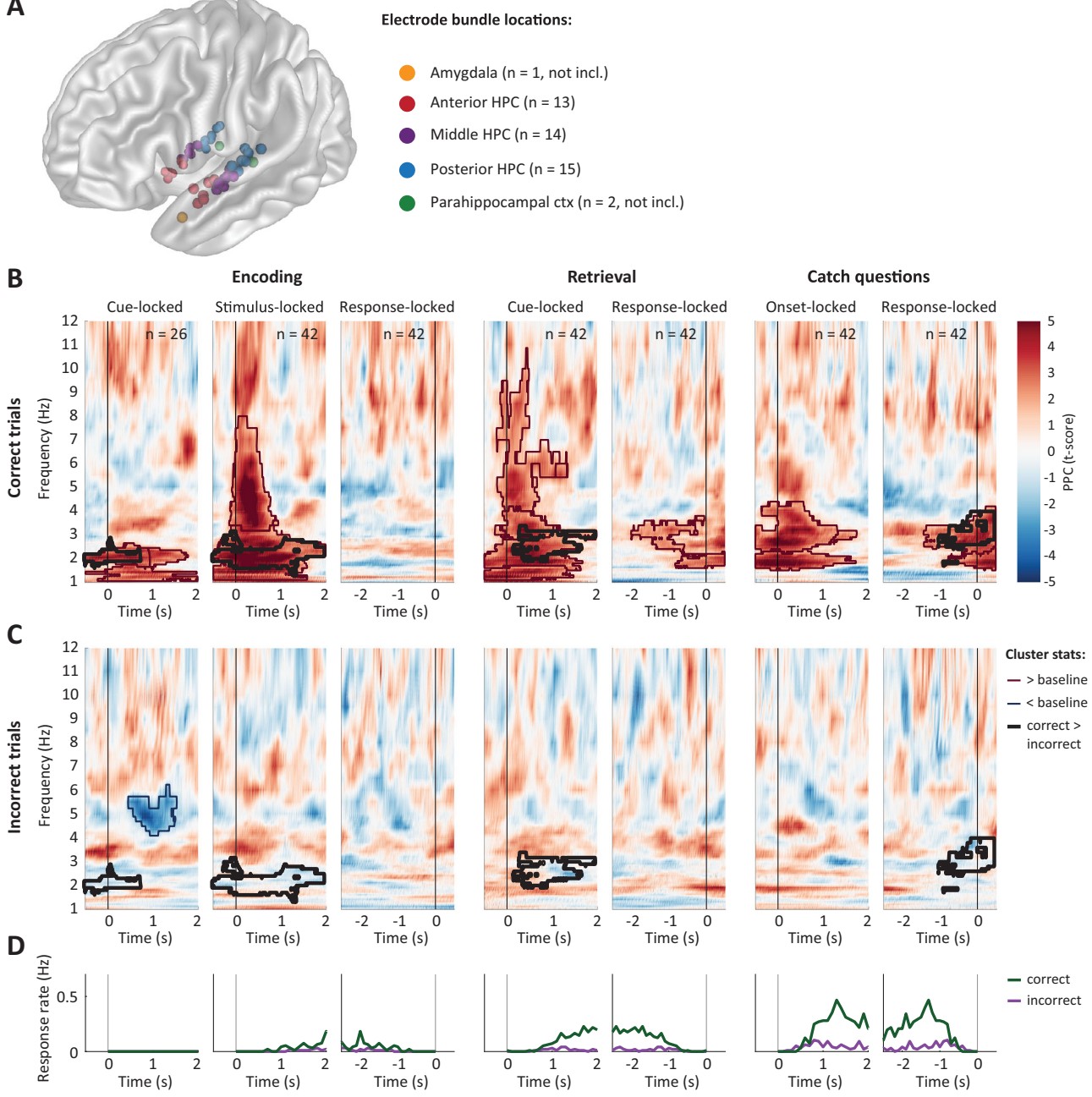

**Fig. 5 Increased phase locking of hippocampal local field potentials during encoding and retrieval. A** Locations of iEEG electrode bundles ($n = 45$, of which $n = 42$ included in further analyses) across all 10 participants, color-coded to indicate five regions of interest (yellow: amygdala; red: anterior hippocampus (HPC); purple: middle HPC; blue: posterior HPC; green: parahippocampal cortex). **B**, **C** Pairwise phase consistency (PPC, color-coded, second-level $t$-score) between correct (**B**) and incorrect (**C**) trials, locked to cue/stimulus onset or response of encoding (left column), retrieval (middle), and catch trials (right). Significant changes from baseline ($\alpha = 0.05$, permutation test against time-shuffled trials) are indicated separately for increases (red) and decreases (blue). Black outlines indicate significant differences between correct and incorrect trials ($\alpha = 0.05$, permutation test against shuffled trial labels, see also Supplementary Fig. 8A). For raw PPC values see Supplementary Fig. 8B and for PPCs per patient see Supplementary Fig. 9. **D** Response rate for correct (green) and incorrect (purple) trials, in the time windows and task phases corresponding to **B** and **C**. Source data are provided as a Source Data file. Number of data points specified in **B** also apply to **C** and **D**.

after cue onset for retrieval trials ($\alpha = 0.05$; cluster-based permutation test against 100 time-shuffled datasets[47]). The clusters of significantly increased PPC (red outlines in Fig. 5B) covered a range of frequencies shortly after cue/stimulus onset, but extended in time in a frequency band between 2 and 3 Hz. This pattern was seen along the long axis of the hippocampus (Supplementary Fig. 10B) and in both hemispheres (Supplementary Fig. 10C), and was also observed for individual patients

(Supplementary Fig. 9), resulting in a high consensus across patients (Supplementary Fig. 10A). The PPC peak frequencies and -values generally aligned with the frequencies and $O$-scores found in the behavioral data of these patients (see Supplementary Fig. 11). This lower theta cluster lasted up to the response (Fig. 5D), and resulted in a significant response-locked PPC cluster for retrieval (encoding showed increased but non-significant response-locked PPC). PPC increases were also visible

in the raw data (Supplementary Fig. 8B) and appeared as theta oscillations in event-related potentials (Supplementary Fig. 8D), confirming that these effects were not caused by changes in baseline. Qualitatively similar PPC increases were found in recordings from hippocampal macro contacts in the same patients (Supplementary Note 1 and Supplementary Fig. 13). Increases in phase consistency were accompanied by increased power during retrieval, but not encoding and catch questions (see Supplementary Fig. 8C), in line with[44], suggesting that amplitude and phase were modulated independently.

In line with the behavioral data, we found no significant increases in PPC for incorrect trials, neither for encoding nor retrieval. When comparing the PPC for correct and incorrect trials within electrodes (cluster-based permutation test against 100 trial-shuffled reference data sets), we found that the PPC increase after cue/stimulus onset in the 2–3 Hz frequency band was significantly stronger for correct than for incorrect trials ($\alpha = 0.05$; black outlines in Fig. 5B, C). The intracranial recordings therefore support our hypothesis that task events induce temporally extended theta phase consistency in hippocampus across correct, but not incorrect trials. By showing that behavioral responses indicating the timing of completed memory encoding and retrieval were preceded by consistent hippocampal theta phases, these findings suggest a potential mechanism for our behavioral findings.

**Encoding and retrieval occur at different phases of the theta rhythm**. The PPC analyses in Fig. 5 demonstrate that theta phases are consistent across trials during both encoding and retrieval phases of the memory task. The identification of phase consistency allows us to ask whether the dominant phases of encoding and retrieval trials differ, which is a prominent suggestion in the computational literature[28]. To this end, we identified the time point and frequency at which PPC was maximal for both stimulus- and response-locked trials during encoding, and cue-locked and response-locked trials for retrieval, for each patient. We then computed the phase differences between encoding and retrieval at the corresponding frequencies and time points for every electrode. Indeed, phase differences between encoding and retrieval trials were non-uniformly distributed around $250.7 \pm 14.1°$ for cue/stimulus-locked trials (Rayleigh's $Z = 30.9$; $p < 0.001$) and differed on average $162.4 \pm 30.6°$ for response-locked trials (Rayleigh's $Z = 7.35$; $p = 0.001$). Both analyses provided support for a half-cycle difference between encoding and retrieval (V-test around 180°; cue/stimulus locked: $V = 33.1$, $p = 0.005$; response-locked: $V = 46.7$, $p = 0.001$; $n = 326$). We tested for inflation of these statistics due to the high channel count (by comparing the V-statistics against 500 time-shuffled datasets) and conclude that phase opposition for the response-locked trials was unlikely to be obtained by chance ($p = 0.042$), while for stimulus/cue-locked data ($p = 0.126$), the observed V-statistic could, in part, be inflated by channel count or a phase bias, for example, due to asymmetry in the theta cycles[48].

To test whether phase opposition generalized beyond the time and frequency with the highest PPC, we computed response-locked event-related potentials for each hippocampal electrode bundle, and filtered these in the theta band (Fig. 6C). We compared the phase of encoding and retrieval ERPs using V-tests in 200 ms sliding windows. After FDR-correction, 55.8% of tested windows supported phase opposition between encoding and retrieval, which is unlikely to be produced by chance. These results further confirm the PPC analyses in Fig. 5 by demonstrating extended phase concentration in the period leading up to and around the button presses, and show that the dominant phases for encoding and retrieval are approximately 180° apart. Together, these results support both theoretical and empirical findings from previous studies that encoding and retrieval processes occur at different phases of the hippocampal theta rhythm, and generalize these findings to LFP recordings from the human hippocampus.

## Discussion

In this study we demonstrated that oscillations can be detected in behavioral responses from associative memory tasks. Using the Oscillation score[43], we showed that button presses that indicate the timing of memory encoding and retrieval were rhythmically modulated, i.e. periodically more or less likely to occur, predominantly in the 1–5 Hz frequency band. We found no evidence for behavioral oscillations for memory-independent task phases. Button presses from forgotten trials did not lock to the oscillation of remembered trials, a distinction that was echoed by hippocampal LFP recordings from 10 epilepsy patients: phase consistency across trials significantly increased in the slow theta range during the encoding and retrieval of later remembered, but not forgotten associations. Finally, phase consistency during encoding and retrieval peaked at opposite phases of the theta cycle, aligning with earlier work suggesting that encoding- and retrieval-related information flows are orchestrated by the phase of the hippocampal theta rhythm. Our data show that these hippocampal mechanisms influence the timing of overt human behavior.

In our study, we relied on button presses that explicitly marked the timing of memory formation and recall. Though these responses are subjective and rely on multiple neural processes, our results allow us to exclude several alternative explanations. Firstly, the behavioral oscillations cannot be explained by rhythmicity in visual processing, as the Encoding and Visual task phases shared identical visual inputs. Secondly, a behavioral oscillation was detectable when memory reinstatement was combined with a catch question (Catch-with-retrieval), but not when the catch question was asked 3 s after reinstatement (Catch-after-retrieval), suggesting that (1) the observed oscillation did not result from motor processes and (2) the lack of oscillations in memory-independent phases cannot be attributed to the nature or content of the catch questions. The data also show that rhythmic clocking is not universal within memory tasks: correct but not incorrect trials showed locking to a theta oscillation, and this result was mirrored in electrophysiology.

We did not observe significant oscillations in behavior for processes that we a priori marked as memory-independent, namely answering catch questions after reinstatement and the visual task. These task phases also did not contain an attentional selection element and did not rely on memory-guided visual search. These cognitive processes were previously linked to theta rhythmic modulation of behavior[37–42] and saccadic eye movements[49–51], respectively. We did however find increased PPC in hippocampal signals after catch questions appeared on screen. O-scores for the corresponding Catch-after-retrieval task phase, although not significant, were higher than for the visual task. Possible explanations are that retrieval-induced oscillations extend in time, or that catch questions induce a second, weaker reinstatement of the memory, leading to behavioral oscillations that are too weak to detect robustly. Alternatively, the oscillations observed for the catch questions could result from maintaining the retrieved object in working memory. Working memory has been proposed to be mediated by theta-nested gamma bursts[52,53], synchronizing a network of cortical memory areas[19,54–57], as well as the hippocampus[58] (for reviews see refs. [59,60]). The micro-electrode recordings presented here do not allow us to distinguish between retrieval- or maintenance-related theta oscillations, and

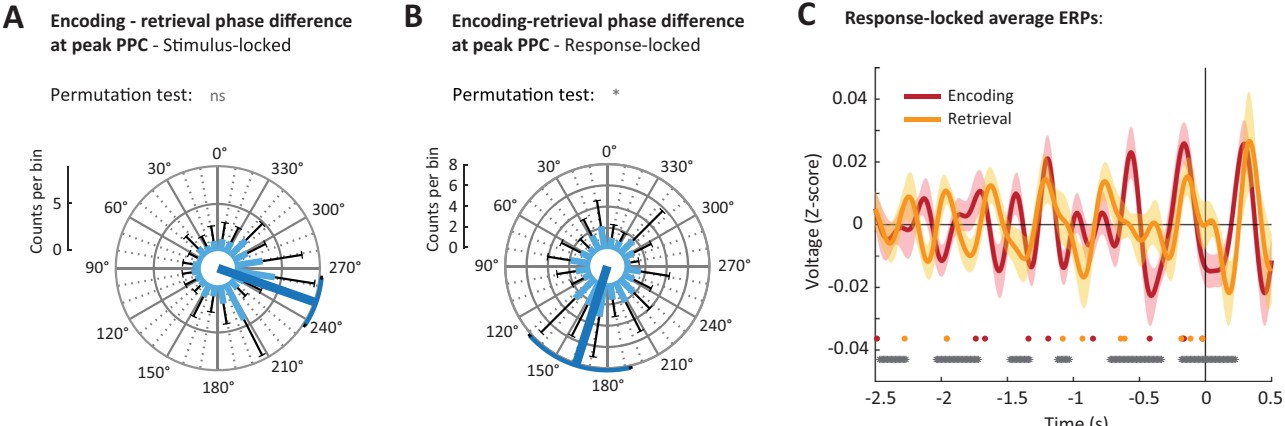

**Fig. 6 Encoding and retrieval occur at different phases of the theta rhythm. A**, **B** Circular histogram of phase differences between encoding and retrieval at the time and frequency of maximal PPC (**A**) following stimulus (encoding) and cue onset (retrieval), and (**B**) before response, across all hippocampal channels ($n = 326$), with the mean direction in dark blue. Light blue bars give the mean and black whiskers the standard deviation across electrode bundles ($n = 42$). V-tests assessed non-uniformity around 180°, and were compared against 500 time-shuffled datasets (permutation test). **C** Event-related potentials (ERPs) for encoding (red) and retrieval (yellow) locked to the patient's response. Solid lines give the mean; shaded areas the SEM. Gray dots indicate time windows with significant phase opposition between encoding and retrieval (V-test in 200 ms sliding windows spaced 10 ms apart; FDR-corrected with $q = 0.05$). Red and yellow dots are the time points of peak PPC for individual patients used in Fig. 6B. Source data are provided as a Source Data file. n.s. not significant; *: $0.05 \geq p > 0.01$; **: $0.01 \geq p > 0.001$; ***: $p \leq 0.001$ (for exact values see the main text).

further work is needed to understand if the behavioral oscillations reported here are specific to long-term memory.

The behavioral theta oscillations for memory-dependent task phases, together with increased PPC in hippocampal LFPs across trials, suggest that events in the memory task (i.e. cue/stimulus onset) induce consistent phase resets in the hippocampal theta rhythm. Our findings suggest this phase reset is most pronounced in the slow 1–5 Hz theta band. Several human intracranial EEG studies have reported prominent slow theta oscillations during episodic memory tasks[7–10], while the higher 4–8 Hz frequency band typically observed in rodents seems to be linked to movement or spatial processing[7]. LFP phase resets and phase locking after task events have been reported for the slow theta band in memory paradigms[16,18,61,62], and phase consistency directly preceding[11] and following[44,45] stimulus presentation has been shown to predict memory performance. In line with our finding, a recent study[44] reported a dissociation between theta power and phase consistency, with power decreasing during encoding, but increasing during retrieval, while phase consistency increased for both processes. Like in rodents, human hippocampal neurons lock their firing to theta oscillations shortly before and during the encoding of later-recognized but not later-forgotten images, for both slow and fast theta bands[18]. In line with our findings, theta phases were found to differ between encoding and retrieval[62], although the effects were limited to an early time window after stimulus presentation, and theta frequencies below 4 Hz were not included in that study. Our intracranial EEG results extend previous findings by demonstrating that post-stimulus phase consistency and encoding-retrieval phase consistency and opposition extend in time in a narrow frequency band, providing a potential neurophysiological mechanism for the theta-clocked behavior.

Our behavioral results align closely with the PPC analyses in terms of dominant frequency and subsequent memory effect; the presence of both LFP phase consistency and behavioral oscillations for correct trials, but absence of both during incorrect trials, suggest a link between the hippocampal rhythms and behavior. Further work is needed to establish how oscillations in hippocampal processes translate to oscillations in behavioral responses. In principle, it is sufficient for hippocampal output to cortical areas to fluctuate rhythmically, i.e., for encoding and recall signals from hippocampus

to occur more frequently at certain time windows. Such fluctuations will then be maintained, though at a delay, in subsequent processing steps that lead up to the motor response, without the necessity for cortical areas to show theta oscillations themselves. Alternatively, behavioral oscillations could arise from theta rhythms in cortical areas that are entrained to or induced by the hippocampal theta rhythm. Coherence with the hippocampus at theta frequencies has been demonstrated for entorhinal[35], parietal[63], and frontal cortices[26] during memory tasks, but it remains to be determined whether hippocampal–cortical theta coherence underlies the behavioral oscillations reported here. Along with optogenetic techniques in rodents[36,64], transcranial magnetic stimulation over lateral parietal cortex in humans might provide a promising way of establishing causality between hippocampal theta and behavior, since it was shown to improve memory performance[65] and hippocampal–cortical coherence particularly when stimulating in theta-bursts[66]. If theta-frequency TMS can enhance memory performance by boosting or entraining theta oscillations, this approach could potentially establish a direct link between hippocampal and behavioral oscillations in healthy humans.

Phase coding is a powerful candidate neural mechanism for optimizing specificity and sensitivity on the one hand, and flexibility on the other. Outside the memory domain, rhythmic switching of visual attention has been demonstrated at theta frequencies[38–41]. In memory tasks, potentially interfering mnemonic information has been shown to recur at different phases[12,26] of the hippocampal theta rhythm. Items kept in working memory are thought to be represented in gamma cycles separated in the theta/alpha phase[52,55]. Visual stimulation at relevant theta/alpha phases, but not opposite phases, boosted working memory performance[67]. Our findings support the notion that not only sensory inputs are sampled periodically by attention, but that internal, mnemonic information is sampled rhythmically as well. Empirical evidence is thus accumulating in both humans and other animals for a powerful role of phase coding and sampling in cognitive processes.

Detecting oscillations in sparse behavioral data is not a trivial task, particularly in memory paradigms that rely on one-shot learning, like the task presented here. The trial counts for these tasks are limited by the number of unique trials participants can perform, which ultimately limits the detectability of oscillations.

We showed that, despite these limitations, the O-score method[43], a method to detect oscillations in spike trains, and our Z-scoring approach were sensitive enough to detect oscillations in behavioral data. Based on simulated datasets, we identified that the sensitivity of the O-score method improved with a higher density of the responses. Interestingly, in our dataset response density was lowest for the encoding task phase, which produced significant O-scores despite the expected reduced sensitivity, suggesting these oscillations are of substantial amplitude. In addition, the simulations showed the O-score method maintained good selectivity in all tested conditions, i.e., did not produce spurious results for weak or absent oscillations, and identified the correct frequency when O-scores were significant. In summary, the O-score method was both sensitive and selective to oscillations for all task phases. It is important to note, however, that a reliable analysis of oscillations in sparse data requires repeated measurements, either in the form of repeated trials[43] or across a large number of participants, like we have done here.

Our results suggest that theta-rhythmicity of memory encoding and retrieval processes can not only be found in neural correlates but also has a clear behavioral signature: the likelihood that a memory is formed or recalled rhythmically fluctuates within a trial, at a slow theta frequency, resulting in rhythmicity of button presses relying on these processes. Our findings suggest that behavior can be a relatively straightforward, yet powerful way to assess rhythmicity of neural memory processes, an approach that can potentially be extended to many other cognitive domains. Together, our behavioral data and hippocampal LFP recordings point to an important mechanistic role for lasting phase consistency in the hippocampal theta rhythm during memory-dependent processing.

## Methods

**Participants**. A total of 216 healthy participants took part in behavioral, EEG and fMRI/EEG studies using the memory tasks described in the next section. A group of 10 epilepsy patients also performed a very similar memory task, more details about this group are given in section "iEEG recordings: patients and recording setup". A separate group of 95 healthy participants completed the visual tasks. All healthy participants volunteered to participate in the studies and were compensated for their time through a cash payment (£6–8 per hour) or the University's course credit system. All participants gave written informed consent before starting the study. None of the healthy participants reported a history of neurological or psychiatric disorders and all had normal or corrected-to-normal vision. Participants only took part in one version of the task, e.g., participants in the behavioral visual task could not take part in the memory EEG study. Only the behavioral data are presented here. A subset of the behavioral data (visual experiment 1 and 2, and memory experiments 5 and 6), as well as the EEG data from experiment 10 (see Supplementary Table 1), were previously reported in ref. [68], while data from experiment 9 were previously reported in ref. [69]. All studies with healthy participants took place in facilities of the University of Birmingham, and the participants were recruited through the university's research participation scheme. All studies were approved by the Science, Technology, Engineering and Mathematics Ethical Review Committee of the University of Birmingham. Demographic information for each of the participant groups is available in Supplementary Table 1.

**Task versions**. In this manuscript we present behavioral and intracranial EEG data recorded during a series of visual and memory experiments. The experiments were originally designed to address the following question: is perceptual information about a stimulus analyzed earlier or later than semantic information, and is this processing order similar when viewing a stimulus compared with reinstating the same stimulus from memory? Data from five experiments (experiments 1, 2, 5, 6, and 10, see Supplementary Table 1) and the analyses addressing the original research question have previously been reported in ref. [68]. Data from experiment 9 (see Supplementary Table 1) were previously reported in ref. [69]. In the present manuscript, we analyze the button presses for perceptual and semantic questions together. We also include the behavioral data from an additional eight follow-up experiments that took place after the collection of the initial datasets.

The experiments can be divided into three main categories (Fig. 1): memory reaction time experiments; electrophysiology memory experiments; and visual reaction time experiments. We give a general description of each category of experiments below, as well as specific differences between experiments within each category. The numbers of participants per task version and their demographic

information is given in Fig. 1 and Supplementary Table 1. The characteristics of each of the 13 task versions are summarized in Supplementary Table 2.

*Groups 1 and 2: Memory experiments*. In the memory experiments, participants first learned associations between cues and objects and later, after a distractor task, memories were reinstated in a cued recall phase, described in more detail below. Participants learned a total of 128 associations, divided into blocks of between four and eight trials. Each block consisted of an encoding phase, a distractor phase, and a retrieval phase. Cues consisted of action verbs (e.g., spin, decorate, hold, …) for all experiments except experiment 12 (details below).

In general, the memory tasks were set up as follows: Each encoding trial started with the presentation of a fixation cross for between 500 and 1500 ms to jitter the onset of the trial. The cue then appeared in the center of the screen for 2 s. After presentation of a fixation cross for 0.5–1.5 s the stimulus (stimuli in experiment 6) appeared. Participants were asked to indicate when they made the association between cue and stimulus by pressing a button (encoding button press). The stimulus remained on the screen for 7 s. After the encoding phase, the participants performed a distractor task in which they judged whether numbers presented on the screen were odd or even. The distractor task lasted 60 s, after which the retrieval phase started. In the retrieval phase the participants were presented with the same cues as during encoding, though in a randomly different order, and asked to recall the associated objects. They then answered either a perceptual or a semantic question about the reinstated object. The trial timed out if the participant did not answer within 10 s. Trials were separated by a fixation cross shown for 500–1500 ms.

The structure of the retrieval phases differed slightly between experiments. We therefore make a further distinction within the memory experiments: the electrophysiology experiments (group 1; experiments 10–13) and the behavioral experiments (group 2; experiments 5–9).

For group 1, we aimed to separate the reinstatement processes from the formulation of the answer to the catch question. To this end, participants were asked to indicate, through a button press, when they had a clear image of the associated object in mind. The trial timed out if the participant did not press the button within 10 s. They then kept the image in mind for 3 s, during which time the screen was blank. Finally, the answer options for the catch question appeared on the screen, after which the participants responded as quickly as possible. Participants had 3 s to respond. As a result, the retrieval trials of the electrophysiology experiments produced two button presses: a retrieval button press and a catch-after-retrieval button press. These button presses are analyzed separately. Only the reinstatement button press is considered memory-dependent, because the catch question appears at a time point when the object has supposedly already been fully retrieved.

For group 2, the answer options were shown on the screen for 3 s before the retrieval cue appeared. The catch-with-retrieval button presses obtained for the memory reaction time experiments can therefore be assumed to represent the time point when sufficient information has been retrieved about the object to answer the catch question.

The number of times we asked participants to retrieve associations was varied between behavioral experiments in group 2. In experiments 5, 7, and 8, every object was probed twice, and participants answered both the perceptual and semantic question for each object in random order. In experiment 9, every object was reinstated six times in the retrieval phase of the block, and twice during a delayed retest 2 days later. The data from the delayed test are not included here due to poor performance (average performance 49.6%, only 16 out of 52 participants performed above chance). In experiment 6, participants learned associations of triplets instead of pairs, consisting of cue, object, and scene image. During the retrieval phase of this experiment, each object was probed only once, and in addition to the perceptual and semantic questions participants were asked a question about the background image (indoor or outdoor?), such that each question was answered on 1/3 of the trials.

The group 1 memory task was used for EEG recordings (experiment 10), combined EEG/fMRI recordings (experiment 11) and intracranial EEG recordings in epilepsy patients (experiments 12 and 13, also see section "iEEG recordings: patients and recording setup"). Several small adjustments were made to the task to accommodate electrophysiology. First, to minimize the duration of the testing sessions, the duration of the distractor phase was reduced to 20 s in the EEG/fMRI experiment, while in the EEG and iEEG task versions, every object was reinstated only once. To compensate for the corresponding drop in the number of catch questions, participants answered both perceptual and semantic catch questions on every trial, one after the other, in random order. The doubling of the number of catch questions per trial was introduced after the first three EEG participants and three iEEG patients were recorded. In addition, the first three iEEG patients learned pairs of background scenes and objects, instead of verb–object pairs, with the background scenes functioning as cues during the retrieval phase (experiment 12). These patients only learned a total of 64 pairs. The reaction time data from these three patients showed no difference to that of the other seven patients (see Supplementary Fig. 1) and no qualitative differences were found in the PPC analysis (PPC per patient is shown in Supplementary Fig. 9A). During encoding trials, the background and object appeared on the screen at the same time. As a result, the encoding trials of these three participants do not have a separate cue period.

We made two further modifications to the task for the iEEG recordings in all epilepsy patients. First, the task was made fully self-paced, such that length of verb

presentation and the period needed to associate cue and object were determined by the patient on each trial. The patients pressed a button when they were ready to move on. Second, to avoid loss of attention/motivation and/or to accommodate medical procedures, visitors, and rest periods, the task was divided into two or three sessions, recorded at different times or on different days. Data from different sessions were pooled and analyzed together. Details of the electrophysiological recordings included in this manuscript can be found in the section "iEEG recordings: patients and recording setup".

*Group 3: Visual reaction time experiments.* In the visual experiments, participants were shown a series of stimuli on the screen, and were asked either a perceptual or a semantic question about each stimulus. The stimuli and questions used in the visual experiments were identical to those in the memory experiments. To get accurate estimates of the reaction times, the answer options were shown for 3 s prior to stimulus presentation. Stimuli were presented in the center of the screen. Each trial was preceded by a fixation cross for a random duration of between 500 and 1500 ms, so the onset of the trial could not be predicted. Like in memory group 2, participants were instructed to answer as fast as possible.

In experiments 1, 3, and 4, all 128 stimuli were shown twice, once followed by a perceptual and once by a semantic question (in random order), so both questions were answered for every object. In experiment 2, in which the object images were shown with a background, all stimuli were presented only once, followed by one of three questions: perceptual, semantic, or contextual, with the later referring to the background (indoor or outdoor). All button presses were included here. We refer to ref. [68] for analyses comparing the different catch questions.

*Stimulus sets.* Across the experiments, three different stimulus sets were used, referred to as Standard, Shape, and Size. Supplementary Table 2 specifies for each experiment which stimulus set was used. Each stimulus set consisted of 128 emotionally neutral, everyday objects. Each object fell into one of two perceptual categories and one of two semantic categories. Participants were instructed about the perceptual and semantic categorizations before onset of the study and were shown examples that were not included in the remainder of the study. In the Standard stimulus set, used in most experiments, the semantic dimension divided the objects into animate and inanimate objects, while in the Shape and Size stimulus sets, used in experiments 3, 4, 7, and 8, objects were categorized as natural or man-made. Furthermore, three different perceptual dimensions were used across the tasks. In the Standard stimulus set, half of the stimuli were colored photographs and the other half were black-and-white drawings. In the Shape and Size stimulus sets only colored photographs were used. Instead, stimuli were categorized as either long or round objects (Shape stimulus set, exp. 3 and 7), or stimuli were presented as large or small pictures on the screen (Size stimulus set, exp. 4 and 8). Stimuli were selected from the BOSS database[70] or other royalty-free online sources.

*Stimulus presentation and pace.* The task presentation was performed using MATLAB 2015a-2018a (The Mathworks Inc.), with Psychophysics Toolbox Version 3 (Releases between January 2017 and April 2019; https://github.com/Psychtoolbox-3/Psychtoolbox-3). With the exception of the fMRI/EEG and iEEG experiments, all experiments took place in dedicated testing rooms at the University of Birmingham, with the participants seated at a desk and watching a computer screen. The computer screens had a refresh rate of 60 Hz. A standard keyboard was used to record the responses. For the fMRI experiment, stimuli were projected onto a screen behind the scanner with a refresh rate of 60 Hz and viewed through a mirror. Participants answered using NATA response boxes. The iEEG experiment was presented and responses were recorded using a laptop (Toshiba Tecra W50) with a screen refresh rate of 60 Hz.

For encoding blocks, trials took on average 9.8 s to complete, resulting in an average of 0.31 visual events per seconds (cue onset, stimulus onset, and stimulus offset). For group 1 retrieval trials took on average 9.0 s for trials with one catch question (0.45 events per second) and 15.2 s for trials with two catch questions (0.39 events per second), while for group 2 retrieval trials took on average 6.8 s (0.44 events per second). Visual task trials took on average 5.8 s, resulting in on average 0.52 events per second. The event rates were below the lower frequency bound of O-score analysis for the behavioral oscillation (minimum of 0.5 Hz, see "RT analysis: O-score and statistics per participant") and were below the 1 Hz lower bound of the PPC analyses, while the screen refresh rate was higher than the upper frequency bound for both the O-score (maximum of 40 Hz) and PPC (12 Hz) procedures. It is therefore not expected that visual events can be the cause of the behavioral oscillations or PPC effects reported here.

**Assessment of performance and exclusion of participants.** Prior to reaction time analyses, the performance of each of the participants was analyzed based on their accuracy in answering the catch questions. Answers to catch question were considered incorrect when subjects chose the wrong answer, when they indicated they had forgotten the answer (for memory tasks) or when they did not answer on time (for healthy participants only). The data of a participant were only included in the analysis of a task phase if the following two requirements were met: (1) catch question accuracy across trials exceeding chance level and (2) a minimum of 10 correct button presses per participant in the task phase of interest. The first criterium was assessed using a one-sided binomial test against a guessing rate of 50% with $\alpha = 0.05$. The

second criterium had to be set as some participants repeatedly failed to provide encoding (reinstatement) button presses before trial time-out, leaving too few trials to run further analyses for the Encoding (Retrieval) phase, despite sufficient performance when answering the catch questions. The inclusion criteria were set a priori. The number of participants included in each of the task phases is shown in Fig. 1 and the number of excluded participants can be found in Supplementary Fig. 1A.

Of the participants who performed the memory tasks, 28 participants answered two catch questions per retrieval trial, while the remaining 198 answered one. To bring the analyses of the 28 participants with 2 catch questions per trial in line with the data from the other 198 participants, we considered a 2-catch trial to be correctly reinstated if one or both catch questions were answered correctly (on average, across 28 subjects: one catch question correct: 12.5% of trials; two catch questions correct: 76.0% of trials, see Supplementary Fig. 1D).

**RT analysis: *O-score* and statistics per participant.** To assess the presence and strength of oscillations in behavioral responses we used the Oscillation score (O-score, Fig. 2B), a method that was developed to analyze oscillations in spike trains[43]. Like spikes, the button presses we study here are discreet, all or nothing events, and can be summarized in trains of button presses across trials. The O-score method identifies the dominant frequency in those trains, and produces a normalized measure of the strength of the oscillation that can be compared across conditions.

The O-score method does not make assumptions about the source underlying the discreet events and can therefore be applied to button presses in a similar way as to spikes, even when the button presses arise from different trials. We did however add an additional processing step before computing the O-score, to compensate for the fact that behavioral responses, unlike spikes, have no baseline rate (e.g., they cannot occur before cue/stimulus onset). Extremely early and late responses therefore have to be considered outliers. We removed these outliers prior to O-score computation by removing the first and last 5% of the button press trace of each participant, i.e., maintaining the middle 90% of the button presses. Supplementary Figure 3B shows that reducing the fraction of button presses included in the analyses affected the ability to identify oscillations, but did not affect the differences found between the task phases.

The button presses from correctly answered trials that remained after outlier removal entered the O-score computation. We made two modifications to the procedure described in ref. [43] to match the characteristics of our dataset. The O-score procedure and our modifications are described below.

The O-score analysis requires the experimenter to define a frequency range of interest. We a priori defined a wide frequency range of interest of between $f_{min}^{init} = 0.5$ and $f_{max}^{init} = 40$ Hz, as we did not want to limit the analyses to a specific frequency band, yet did not expect to detect frequencies in the higher gamma frequency band. Given the wide variety in the number of responses and response times, we checked for every participant and task phase whether these pre-set frequency bounds were valid. Following ref. [43], we increased the lower bound to $1/c_{min}$ of the width of the response distribution (in seconds) of the participant, with $c_{min} = 3$, such that at least three cycles of the lowest detectable frequency were present in the data. We reduced the upper bound to the average response rate (button presses per second), if the participant did not have enough button presses to resolve the upper frequency limit. The O-score was then computed through the following series of steps:

Step 1: As described in ref. [43], we computed the ACH of the button presses with a time bin size of 1 ms ($f_s = 1000$ Hz).

Step 2: The ACH was smoothed with a Gaussian kernel with a standard deviation $\sigma_{fast}$ of 2 ms. As estimated in ref. [43], this smoothing kernel attenuated frequencies up to 67 Hz by less than 3 dB, allowing us to detect frequencies in the entire frequency range of interest.

Step 3: We identified the width of the peak in the ACH using the method described in ref. [43]. However, to avoid the introduction of low frequencies by replacing the peak, we opted to only use positive lags beyond the detected ACH peak for further steps, as the peak-replacement approach would not allow us to detect frequencies toward the lower bound of our frequency range of interest. To identify the peak, we smoothed the ACH with a Gaussian kernel with $\sigma_{slow}$ of 8 ms, resulting in the smoothed ACH trace $A_{slow}(l)$, with $l$ the lag. We then identified the left boundary lag of the central peak $l_{left}$ by

$$l_{left} = l \left| \Delta A_{slow}(l) \frac{2\, l_{max} + 1}{A_{slow}(0)} \le \tan\left(\frac{10\,\pi}{180}\right) \right. \tag{1}$$

where $l_{max}$ is the highest lag included in the ACH and $A_{slow}(0)$ is the value of the peak of the ACH (i.e. at lag 0).

Step 4: The remaining part of the ACH was subsequently truncated/zero padded to size $w$, where

$$w = 2^{\left\lfloor \max\left( \log_2\left( 2c_{min}\frac{f_s}{f_{min}} \right), \log_2\left( \frac{l_s}{2} \right) \right) \right\rfloor + 1} \tag{2}$$

We then applied a Hanning taper and the Fourier transform was computed.

Step 5: We identified the frequency with the highest power in the participant-adjusted frequency bounds, as well as the average magnitude of the spectrum

between 0 and $f_s/2$ Hz. The $O$-score was then computed as

$$O = \frac{M_{peak}}{M_{avg}} \quad (3)$$

In their paper, Muresan and colleagues propose a method to estimate the confidence interval of the $O$-score, allowing for a statistical assessment at the single cell level. However, this approach requires multiple repeated recordings, which are not available for the data presented here, nor do the datasets contain enough data points to create independent folds. Instead, we opted to generate a participant-specific reference distribution of $O$-scores for the identified frequency, to which we could compare the observed $O$-score. To this end, we randomly generated 500 time series for each participant matching the trial count and overall response density function of the participant's original button presses. First, a gamma probability function $r_{gamma}(t)$ was fitted to the participant's response distribution using the fitdist function from the Statistics and Machine Learning Toolbox (v11.3) for MATLAB 2018a (The Mathworks Inc.; for an example see the gray lines in Fig. 2), and scaled to the number of responses of the participant. We then generated 500 Poisson time series, with the probability of a response in a time step $\triangle t = 0.5$ ms, given by

$$P_{resp}(t \rightarrow t + \triangle t) = r_{gamma}(t)\triangle t \quad (4)$$

If a gamma distribution could not be fitted (as assessed through a $\chi^2$ goodness-of-fit test with $\alpha = 0.05$), the participant's button presses were instead randomly redistributed in time, with the new time per button press uniformly drawn from a window defined by one period of the participant's peak frequency and centered around the time of the original button press. Redistributing within one period of the identified oscillation ensured that this oscillation frequency of interest was not maintained in the reference data set, while minimizing changes to the overall response distribution.

$O$-scores were then computed for each of the resulting reference traces, but instead of finding the peak, the power at the peak frequency of the observed $O$-score was used. This approach controls for any frequency bias that could arise due to the length of the time series and/or the number of data points included in the analysis. To compare the observed $O$-score to the reference $O$-scores, we first log-transform all $O$-score values. This log-transformation was needed as the $O$-score is a bounded measure (it cannot take values below 0) and the $O$-score distribution is therefore right-skewed when $O$-score values are low, leading to an underestimation of the standard deviation of the reference distribution. The log-transformed reference $O$-scores were then used to perform a one-tailed $Z$-test for the observed $O$-score at $\alpha = 0.05$, establishing the significance of the oscillation at the single participant level. For a validation that 500 reference $O$-scores was sufficient to produce a stable outcome for the $Z$-scoring, we refer to Supplementary Fig. 3A. Second-level $t$-scores were subsequently computed based on the $Z$-scored $O$-scores for each task phase and tested with $\alpha = 0.01$ (one-tailed, Bonferroni-corrected for 5 task phases). These $Z$-scored oscillation scores can be assumed to represent the strength of the behavioral oscillation, and are the basis of many of our statistical comparisons.

To test whether the $O$-scores of memory-dependent task phases, i.e., Encoding, Retrieval. and Catch-with-retrieval, and memory-independent task phases, i.e., Catch-after-retrieval and Visual, against each other, we fitted a linear mixed model to the $Z$-scored $O$-scores, with memory dependence and the length of the time series used for $O$-score computation as fixed effects, and an intercept per participant as random effect. We included the length of the time series, computed as the difference (in seconds) between the last and the first RT used in the $O$-score analysis, because there was a substantial difference in response times between the task phases, with overall similar patterns as the $O$-scores (see Supplementary Fig. 1). We included participants as random effects to compensate for the difference in the number of data points contributed by memory task participants (3 data points from group 1, 2 data points from group 2) compared to visual task participants (1 datapoint from group 3), and to account for dependencies in the data. We fitted an identical linear mixed model to the peak frequencies corresponding to significant $O$-scores. The linear mixed models were fitted using the fitlme function from the Statistics and Machine Learning Toolbox (v11.3) for MATLAB 2018a (The Mathworks Inc.).

The performance of the modified $O$-score method and $Z$-scoring procedure were tested in a simulated dataset where the amplitude and frequency of the oscillation in the simulated button presses was varied. Methods and results of these simulations are given in Supplementary Note 2.

**RT analysis: phase of response.** For the task phases with significant second-level $O$-scores, i.e., Encoding, Retrieval, and Catch-with-retrieval, we analyzed the phases at which individual button presses occurred in the behavioral oscillation identified by the $O$-score analysis. We performed this analysis for both correctly and incorrectly remembered trials. As this analysis relied on the frequency identified by the $O$-score analysis, only participants with significant $O$-scores were included.

To identify the phases of the button presses, we first established a continuous reference trace that captured the behavioral oscillation. This was achieved by convolving the button presses with a Gaussian kernel, with $\sigma_{freq} = f_{peak}/8$. The resulting continuous trace was then band-pass filtered with second-order Butterworth filter with a 1 Hz wide pass band centered on the participant's peak

frequency identified by the $O$-score. The filtered trace was then Hilbert transformed and the instantaneous phase was computed, resulting in a phase of 0 rad for the peak of the behavioral oscillation. Finally, for each button press, the corresponding phase of the reference trace was determined and stored for further analyses.

We used two complementary approaches to compare the phase locking of correct versus incorrect trials: across participants, allowing us to include correct and incorrect trials from all participants with significant $O$-scores, even when the number of incorrect button presses was low; and within participants, comparing the phase distributions of correct and incorrect trials for participants with 10 or more incorrect trials. These approaches are described in more detail below.

With the across-participant analysis we aimed to address the following questions: (1) are correct and incorrect trials phase-locked to the behavioral oscillation found for the correct trials and (2) are correct trials locked to this oscillation more strongly than incorrect trials? For these analyses, to find the phases of incorrect trials, we compared the timing of the incorrect button presses to the phase trace determined on the correct trials only. To determine the phases of the correct trials, to avoid circularity, we instead used a leave-one-out approach; for each correct button press, a phase trace was established based on all other correct trials. We then performed a $V$-test (implementation: CircStats toolbox 2012a[71]; https://github.com/circstat/circstat-matlab) to assess non-uniformity of the phase distributions around the peak of the behavioral oscillation (i.e. around phase 0 rad), providing an answer to the first question. To address the second question, i.e., whether correct phase distributions were modulated more strongly than incorrect phase distributions, we had to compensate for the trial count differences as well as the methodological differences in determining the phase distributions for correct and incorrect trials. To this end, we defined the permutation test statistic:

$$V_{diff} = V_{correct} - V_{incorrect} \quad (5)$$

with $V$ being the test statistic from the $V$-test for non-uniformity around phase 0. For each participant with a significant $O$-score, we then randomly shuffled the labels of the correct and incorrect trials, and computed the $V_{diff}$ statistic across participants for the label-shuffled trials in the same way as described for the observed labels. We repeated this shuffling procedure 100 times and counted the number of times $V_{diff}^{observed}$ was smaller than $V_{diff}^{shuffled}$. This procedure hence resulted in a $p$ value that estimated the likelihood that the observed difference in phase modulation between correct and incorrect trials was produced by chance.

For participants with sufficient (10 or more) incorrect trials, we performed an additional analysis to compare the phase modulation of correct and incorrect trials. For these participants, the correct trials were randomly subsampled to match the number of incorrect trials. The phases of the incorrect trials and the subsampled correct trials were then determined based on the phase trace of the remaining correct trials and $V$-tests were performed for both subsampled correct and incorrect phase distributions. The $V$-statistics for correct trials were then compared to those for incorrect trials using paired $t$-tests. The subsampling procedure was repeated 100 times.

**iEEG recordings: patients and recording setup.** We recorded intracranial EEG from 10 epilepsy patients while they were admitted to hospital for assessment for focus resection surgery; 7 patients were recorded in the Queen Elizabeth Hospital Birmingham (Birmingham, UK) and 3 patients in the Universitätsklinikum Erlangen (Erlangen, Germany). For an 11th patient, task recording was aborted due to poor performance. All patients were recruited by the clinical team, were informed about the study and gave written informed consent before their stay in hospital. Ethical approval was granted by the National Health Service Health Research Authority (15/WM/2019), the Research Governance & Ethics Committee from the University of Birmingham, and the Ethik-Kommission der Friedrich-Alexander Universität Erlangen-Nürnberg (142_12 B).

As part of their routine clinical care, the patients were implanted with intracranial depth electrodes targeting the medial temporal lobe, as well as other brain areas. Patients gave written informed consent for the implantation of between two and eight Behnke-Fried electrodes with microwire bundles (AdTech Medical Instrument Corporation, USA) in the medial temporal lobe (see Fig. 1 for electrode placement and Supplementary Table 7 for electrode numbers per patient). Only data from the hippocampal electrodes are presented here. Implantation schemes were determined by the clinical team and were based solely on clinical requirements. Each microwire bundle contained eight high-impedance wires and one low impedance wire, which was used as reference in most patients (see Supplementary Table 7 for patient-specific references). Data were recorded using an ATLAS recording setup (Neuralynx Inc, USA.) consisting of CHET-10-A pre-amplifiers and a Digital Lynx NX amplifier and running on the Cheetah software version 1.1.0. Data were filtered using analog filters with cut-off frequencies at 0.1 and 9000 Hz (40 Hz for patient 01) and sampled at 32,000 Hz in Birmingham and 32,768 Hz in Erlangen. All data were stored on the CaStLeS storage facility of the University of Birmingham[72].

For each patient both pre- and post-surgical T1-weighted MRI images were acquired. The pre- and post-surgical scans were co-registered and normalized to MNI space using SPM12 (https://www.fil.ion.ucl.ac.uk/spm/). The locations of the tip of the macro-electrodes were determined through visual inspection using MRIcron (v1.0.20190902; https://people.cas.sc.edu/rorden/mricron/index.html) and electrodes were assigned one of the following anatomical labels: amygdala,

anterior, middle or posterior hippocampus, or parahippocampal gyrus. The locations and labels were visualized using ModelGUI (release 1.0.30; http://www.modelgui.org) and are shown in Fig. 5A.

The patients performed the memory task described in section "Task versions" on a laptop computer (Toshiba Tecra W50), while seated in their hospital bed or on a chair next to their bed. The three patients who were recorded in Erlangen, Germany, performed the task in German. Patients completed between 64 and 128 full trials, divided over between 1 and 3 recording sessions (see Supplementary Table 8). Of the 10 patients, 3 patients performed a version of the task that used scene images as cue (see "Task versions"), while the other patients were presented with verbs as cues. For the image cue task version, the cue was shown at the same time as the object; hence, the encoding data of three patients had no separate cue phase.

**iEEG analysis: LFP data preprocessing**. Raw microwire data were loaded into MATLAB 2018a using the MatlabImportExport scripts (version 6.0.0: https://neuralynx.com/software/category/matlab-netcom-utilities) provided by Neuralynx Inc. The data were subsequently zero-phase filtered with a third-order FIR high-pass filter with a cut-off frequency of 0.5 Hz and a sixth-order FIR low-pass filter with a cut-off frequency of 200 Hz using FieldTrip (v20190615 (ref. [73]); https://github.com/fieldtrip/fieldtrip). A Notch filter with a stopband of 0.5 Hz wide at −3 dB was used to remove 50 Hz line noise and its harmonics. The data were down-sampled to 1000 Hz and divided into encoding and retrieval trials.

All data were visually inspected and channels/time points that contained electrical artefacts or epileptic activity were removed. Trials that had more than 20% of time points marked as artefactual were rejected in their entirety. In an additional preprocessing step, the data of patient 03 were re-referenced against the mean of the channels in each microwire bundle. This was done to bring the data from this patient, whose data were originally recorded against ground, more in line with the referencing schemes of the other patients, which were recorded against a local reference wire (see Supplementary Table 7 for reference information per patient).

**iEEG analysis: wavelet transform, pairwise phase consistency and cluster statistics**. The pre-processed microwire recordings were wavelet transformed using a complex Morlet wavelet with a bandwidth parameter of 4. We used the cwt implementation from the Wavelet Toolbox (v5.0) for MATLAB 2018a (The Mathworks Inc., USA) to compute the wavelet transform. The wavelet was scaled to cover a frequency range between 1 and 12 Hz in 43 pseudo-logarithmic steps and convolved with the data in time steps of 10 ms.

To obtain the power plots in Supplementary Fig. 8C, we extracted the absolute value of the wavelet coefficients and assessed power changes per frequency against a −2 to −0.5 s pre-cue baseline using a two-sided $t$-test for every time point. We averaged the resulting t-maps across the wires within each bundle, as they shared a common low impedance reference. The bundle averages were then used to compute a second level $t$-score across the bundles of all participants. We performed second-level analyses at the level of bundles, because correlations between signals from two bundles from the same patient were low and did not differ from correlations between signals from two bundles from two different patients, suggesting bundle was the main source of variance (see Supplementary Fig. 10A). The $p$ values resulting from the second-level analysis were entered into a Benjamini–Hochberg false discovery rate (FDR) correction procedure with $q = 0.05$ to correct for multiple comparisons and the $t$-score map was masked at alpha = 0.05.

The phases obtained for every frequency and time point in the trial using the complex wavelet transform were used to compute the pairwise phase consistency (PPC[46]) across trials for each time- and frequency pixel and for each microwire. The PPC was calculated for correct and incorrect trials separately. The PPC values were then non-parametrically tested relative to their pre-cue baseline, defined as the period from 2 to 0.5 s prior to cue onset, using a Mann–Whitney $U$-test. We opted for a non-parametric test due to the strong left-skew of the PPC data. As for the power analyses, the resulting approximated $Z$-values were averaged across the microwires in a bundle, and the averages were used to compute a second level $t$-score across all bundles from all patients.

We then detected time–frequency clusters of significant PPC through the following steps. First, the $t$-scored PPC values were thresholded at $\alpha = 0.05$ with df = $N_{bundles}$ − 1, resulting in a binary image with 0 = non-significant and 1 = significant. This binary image was entered into an 8-connected component labeling algorithm to identify clusters of significant PPC values.

As we used a fixed threshold to identify the clusters, it is possible for clusters to be made up of two or more merged peaks. This merging of peaks artificially inflates the cluster's size. To avoid this, we tested whether each cluster contained more than one peak, and if so, split the cluster. To this end, for every cluster, we iteratively increased the significance threshold towards 90% of the highest value in the cluster, in 5% increments, and reran the cluster detection method described in the previous paragraph. We required any resulting subclusters to be at least 5% of the size of the original cluster, to overcome noise in the data. If no subclusters were found, the threshold was increased further. On the other hand, if all identified subclusters were smaller than 5% of the original cluster, we concluded that the cluster could not be split. If subclusters of sufficient size were detected, these were stored. For all

pixels that were part of the original cluster, but were not a member of any of the new subclusters, we computed the weighted Euclidian distance to all subclusters and assigned them to the closest subcluster. For each resulting (sub)cluster we then computed a cluster statistic defined as the sum of all $t$-scores from all pixels in the cluster.

We took a non-parametric approach to assess the statistics at the cluster level. To this end, we went back to the wavelet transforms and, at a random time point in each trial, divided the trial in two parts. We then concatenated the first part of the trial to the end of the second part. This procedure, suggested in ref. [74], left all characteristics of the dataset intact, with the exception of the temporal structure of the phase. We computed the PPC across these time-shuffled trials, $Z$-scored against baseline, computed the second level $t$-score, identified clusters of significant $t$-scores, and computed the cluster scores as described in the previous paragraph. We repeated this procedure 100 times and we stored the highest cluster score for each repetition, resulting in a reference distribution of maximum cluster scores. We then non-parametrically compared the cluster scores from the intact data to the reference distribution, with $\alpha = 0.05$. We performed the time-shuffle analysis independently for positive and negative changes in PPC and for correct and incorrect trials separately.

Finally, we also compared the PPCs from correct and incorrect trials to each other directly. We used a similar approach as described above, with two important differences: (1) the second-level analysis was now performed on the pairwise difference between correct and incorrect PPCs from the same bundle and (2) we shuffled correct and incorrect trials (as opposed to time points) to obtain the reference distribution.

**Phase differences and event-related potentials**. We used two different approaches to assess whether phases between encoding and retrieval trials differed. First, we tested whether encoding and retrieval phases differed at the moment of peak PPC, i.e., where the effect of phase resets was optimal and trials were most phase aligned. To this end, we detected the highest PPC value for every participant and stored the average phase for every electrode at the corresponding time and frequency. We then computed the phase difference per electrode by subtracting the retrieval phases from the encoding phases. This procedure was performed on both the cue- (for retrieval) or stimulus- (for encoding) locked data and for the response-locked data. We subsequently performed $V$-tests for non-uniformity around 180° (CircStats toolbox 2012a (ref. [71]); https://github.com/circstat/circstat-matlab) to assess whether the phases of encoding and retrieval were opposite at peak PPC. We compared the $V$-statistics to $V$-statistics computed using the same approach in 500 time-shuffled datasets (see previous section).

For the second approach we computed event-related potentials (ERPs) to test for phase opposition in time windows leading up to the response. To obtain the ERPs, we first $Z$-scored the raw data per electrode by subtracting the mean and dividing by the standard deviation of all trials and time points. We then tested whether all electrodes had the same sign. This step was essential because recordings from different layers of the hippocampus can have opposing polarities. In microwire recordings there is no control over the placement of the electrode, nor is it possible to determine this placement based on scans, hence potential sign flips have to be detected in the data, before averaging data of different electrodes. We detected the sign by identifying the highest deflection in the trial-average of every electrode in the 1 s time interval after cue onset during encoding. If this deflection was negative, the data from the electrode was flipped. Note that the time interval we used for sign testing was not included in the ERP analysis in Fig. 6. We then averaged the trials of all wires within a microwire bundle, separating correct and incorrect trials. The data resulting from this step are represented in Supplementary Fig. 8D. We then filtered the averaged data in the theta-frequency band (1–5 Hz; data in Fig. 6C) and identified the instantaneous phase using the Hilbert transform. We subtracted the instantaneous phases from the retrieval trials of the phases from the encoding trials for each bundle yielding the instantaneous phase difference. The phase differences were collected in windows of 200 ms (i.e., 1 cycle at the 5 Hz upper bound of the theta band) spaced 10 ms apart and we tested whether the phase differences in each window were non-uniformly distributed around 180° using a $V$-test (CircStats toolbox 2012a (ref. [71]): https://github.com/circstat/circstat-matlab). We used a Benjamini–Hochberg false discovery rate correction procedure[75] with $q = 0.05$ to account for repeated tests across the time windows.

**Reporting summary**. Further information on experimental design is available in the Nature Research Reporting Summary linked to this paper.

## Data availability

All behavioral data underlying the results in this study and from the iEEG dataset, the PPC values for correct and incorrect trials, including the time- and trial-shuffled PPCs have been deposited in the following FigShare repository: https://doi.org/10.6084/m9.figshare.c.5192567 (ref. [76]). Behavioral data from experiments 1, 2, 5, 6, and 10 (see Supplementary Table 1) were previously reported in ref. [68]. Data from experiment 9 (see Supplementary Table 1) were previously reported in ref. [69]. Intermediate processing steps and other derived iEEG data will be made available upon reasonable request. The raw iEEG data and patient-specific electrode locations are protected and are not available due to data privacy laws. Source data are provided with this paper.

## Code availability

Custom MATLAB functions and scripts used to produce the results presented in this study are publicly available via GitHub: https://github.com/marijeterwal/behavioral-oscillations and FigShare: https://doi.org/10.6084/m9.figshare.13213769 (ref. [77]).

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

## Acknowledgements

This work was funded by starting grant ERC-2016-STG-715714 (STREAM) of the European Research Council to M.W., consolidator grant ERC-2015-647954 (Code4-Memory) awarded to S.H., and a Wellcome Trust/Royal Society Sir Henry Dale Fellowship (10762/Z/15/Z) awarded to B.S. We thank Sophie Watson, Wing Tse, Jonathan Burton-Barr, Emma Sutton, Thomas Faherty, Alexandru-Andrei Moise, Laura De Herde, Britanny Lowe, Jessica Davies, and James Lloyd-Cox for their help with collecting the behavioral data and Andrew Reid, Gernot Kreiselmeyer, and Rüdiger Hopfengärtner for technical support. We are grateful to all participants for donating their time, and in particular thank the patients, their families, and the hospital staff for accommodating our work.

## Author contributions

J.L.-D., J. Lifanov, and M.W. designed the experiments, and M.t.W., J.L.-D., J. Lifanov, F.R., L.D.K., S.G., J. Lang, H.H., D.R., V.S., R.C., B.S., S.H., and M.W. were involved in data collection. M.t.W. performed the data analysis and M.t.W. and M.W. wrote the manuscript. All authors provided feedback on the manuscript.

## Competing interests

The authors declare no competing interests.
