## [Peer Review File · Nature Communications]

Theta rhythmicity governs human behavior and hippocampal signals during memory-dependent tasksREVIEWER COMMENTS

Reviewer #1 (Remarks to the Author):

The authors present evidence for a hippocampal phase reset effect that modulates memory retrieval using two distinct analyses. In the first, they use the response times on a source memory task to fit extract an “oscillation” by applying an FFT to the “spike train” of the timing of correct responses across trials relative to the cue onset. This is done using the O score, a straightforward method that uses power at each frequency relative to power across the spectrum. This identifies a larger O score for oscillations in the 1-5 Hz frequency range. This means that responses across trials tend to occur either at lags of 1 second, 2 seconds, 3 seconds etc (for a 1 Hz oscillation) or 1.5 sec, 2 sec, 2.5 sec etc for a 2 Hz oscillation, from what I can gather. This is a highly novel idea, and I think it will be a good addition to the literature.

The original application of the O score utilizes continuous spike trains in time, whereas this application uses the timing of responses compiled across trials. This sets a fairly high bar to convince readers regarding application of the method, but the authors perform a series of control analyses to overcome this problem that are helpful. Having said that - the incorrect response trials show a distribution with greater variance than correct responses. Does this have an impact on the application of O score?

The impact of the finding depends on how strongly the readers are convinced that there is periodicity in the underlying behavioral responses. The example plot in Figure 2 is helpful. I would like to see such plots for other participants, especially since the distribution of behavioral response times in this patient seem to be uncharacteristic of the overall times shown in the supplemental material. Another step that might help overcome skepticism is to fit the resulting oscillation from the FFT extraction to the behavioral histogram and show that the periodicity lines up with the behavior (perhaps with a goodness of fit test). I understand the analysis in Figure 4 addresses this point to some degree, although the degrees of freedom are quite high for the behavioral effect due to combining all trials into a single test (I think).

The hardest part of this method to wrap my head around is the use of the autocorrelation result to fit the FFT. Applying autocorrelation to a spike train that is essentially a continuous time series makes sense, but the behavioral responses are discontinuous in time. The result in Fig 2A makes it look like you can run the FFT on the response histograms themselves, which would be more convincing.

The control analysis applied to visual recognition has a very different distribution of response times that are smaller and more highly constrained, so I am not sure it is an effective control. Additional detail regarding how the shuffle distributions were constructed for the correct trials would also be helpful.

The second half of the paper looks at LFPs from BF microwires, showing a phase reset effect. This result has been demonstrated during associative memory and other paradigms, including a success effect. These results add additional support to these previous findings, and certainly phase reset is a necessary condition for their behavioral analysis to work. While these findings are interesting, they occur in a relatively small group of patients - does the success effect for phase result reflect random effects analysis? Do these same 10 subjects exhibit periodicity in their behavioral responses that matches the frequency of maximal phase reset? That would be especially interesting.

Reviewer #2 (Remarks to the Author):

This manuscript investigates theta oscillations and their behavioral relevance in human memory tasks using a clever behavioral analysis along with intracerebral recordings. The authors analyzed button responses during both memory and visual perception tasks using the O score, a method adapted from spike train analysis, and found evidence for slow-theta rhythmicity in these responses only

during memory tasks. Correct memory was associated with hippocampal phase consistency in this slow theta band for correct trials. Overall, this manuscript is clearly written, well-powered, and uses appropriate methods to test novel predictions about theta oscillations. I did not find too many weaknesses and believe this manuscript will be impactful in the field. Nonetheless, I provide below several points which could improve the manuscript and make it suitable for publication.

Major comments

1. The authors report neural results from micro LFP contacts and this is well justified in my view. On the other hand, the majority of human iEEG studies have analyzed data from macro contacts. It would therefore be helpful to readers to include results from the macro contacts as a supplemental analysis. Do results differ between macro and micro-contacts?
2. Human theta can show substantial variability between subjects and channels. Was there relationship between the frequency of O score and frequency of peak PPC across subjects? Similarly, did the authors observe any anterior-posterior or laterality differences in PPC?
3. The authors found no effects in visual tasks. On one hand, this is a nice demonstration of the memory-dependence of their effects, but another line of literature has shown modulation of hippocampal LFPs with eye movements (e.g. Jutras et al., Kragel et al., Hoffman et al.). I suggest the authors include some description of the key task differences between studies in the discussion.
4. I am concerned by the high channel count (n=326 channels, Figure 6 caption; Table S6) from a small number of patients. It is doubtful that the signals on channels are independent and therefore the degrees of freedom are likely over-inflated.

Minor comments

5. Is zero radians the peak or trough in the figures? Please add these labels, as different groups use different conventions (unfortunately!).
6. Did the controls and patients differ on O score? Could these methods be used to identify MTL dysfunction in epilepsy patients in lieu of intracranial monitoring?

Response to the reviewers

Summary of the changes:

We thank the reviewers for their thoughtful and constructive comments on the earlier version of our manuscript. In response to the feedback provided by the reviewers, we have performed several new analyses, most notably: 1) we pre-processed the macro data from the patients and computed phase coherence for the macro contacts in hippocampus, replicating the results from the microwires; 2) we performed additional statistical analyses on both the behavioral data and the microwire PPC data, corroborating our initial claims; and 3) ran some additional simulations to test and validate the O-score method, confirming that it does not produce spurious oscillations when used on this type of behavioural data. The comments have also led to several clarifications in the figures of the main text and in the Methods section, and to several new pointers in the Discussion. The following new figures were added to the Supplementary Information: Figure S2, S6 H and I, S8 and S9, showing more data and analyses for both behavior and electrophysiology. We decided to include these new results (e.g., the PPC on the macro contacts) in the supplements rather than the main results, in order to not further add to the already high density of information and figures. Having said that, we feel that the new data and analyses have substantially increased the confidence in the results presented in the main text. Please find a detailed point-by-point response to each of the comments below.

Reviewer #1:

'The authors present evidence for a hippocampal phase reset effect that modulates memory retrieval using two distinct analyses. In the first, they use the response times on a source memory task to fit extract an "oscillation" by applying an FFT to the "spike train" of the timing of correct responses across trials relative to the cue onset. This is done using the O score, a straightforward method that uses power at each frequency relative to power across the spectrum. This identifies a larger O score for oscillations in the 1-5 Hz frequency range. This means that responses across trials tend to occur either at lags of 1 second, 2 seconds, 3 seconds etc (for a 1 Hz oscillation) or 1.5 sec, 2 sec, 2.5 sec etc for a 2 Hz oscillation, from what I can gather. This is a highly novel idea, and I think it will be a good addition to the literature.'

1A. *'The original application of the O score utilizes continuous spike trains in time, whereas this application uses the timing of responses compiled across trials. This sets a fairly high bar to convince readers regarding application of the method, but the authors perform a series of control analyses to overcome this problem that are helpful. Having said that - the incorrect response trials show a distribution with greater variance than correct responses. Does this have an impact on the application of O score?'*

We agree with the reviewer that performance of the O-score method is an important factor to assess, especially when comparing datasets with different numbers of data points and/or different distributions. This was a prominent reason for performing the simulations described in the SI (sections 5 and 6). We would like to point out that we did not perform simulations for the incorrect trials or discuss the possible impact of high variance on the correct/incorrect comparison, because we did not use the O-score analysis on the incorrect trials in the original dataset. The reason for this was that a substantial part of the participants did not reach our *a priori* inclusion criterium for the O-score analysis of 10 incorrect trials. We therefore designed the analyses described in the section 'Reaction times of incorrect trials are not locked to the behavioral oscillation' to compare the correct and incorrect trials independent of the O-score method.

Our simulations do give some interesting insight into the possible effect of small dataset/large variance on the O-score method. Due to the task design, the encoding phase of the task has the lowest number of correct responses per participant, while these responses are spread out over the largest time window (see Table S3 and Figure S1C). As a result, the encoding task phase has a substantially lower number of responses per second than the other task phases. Our simulations suggest that this low response density reduced the **sensitivity** of the O-score method. For the encoding task phase, the O-score could only detect stronger oscillations; the O-score produces significant results only for oscillation amplitudes of 60% and higher for encoding, while for retrieval and visual task phases it produces significant results at 30-40 % (see Figure S10B-D). However, importantly, the low response density for encoding trials did not affect the **selectivity** of the method, i.e. the O-score method did not produce spurious significant results for weak or absent oscillations, and where it produced a significant result, it did so for the correct frequency (see Figure S10B). This suggests that indeed, a direct comparison of O-scores should not be performed without control/correction (such as downsampling) on datasets with unequal response densities. On the other hand, it strengthens our finding of oscillations in the encoding task phase, suggesting these oscillations must be of substantial amplitude.

To aid the understanding of the performance of the O-score we have added a description of the impact of response density on sensitivity and selectivity of the O-score to the Discussion section:

'Based on simulated datasets, we identified that the sensitivity of the O-score method improved with a higher density of the responses in time. Interestingly, in our dataset the response density was lowest for the encoding phase of the task, which produced significant O-scores despite the expected reduced sensitivity, suggesting these oscillations are of substantial amplitude. In addition, the simulations showed the O-score method maintained good selectivity in all tested conditions, i.e. did not produce spurious results for weak or absent oscillations, and identified the correct frequency when O-scores were significant. In summary, the O-score method was both sensitive and selective to oscillations for all task phases.' (lines 517-524)

1B. *'The impact of the finding depends on how strongly the readers are convinced that there is periodicity in the underlying behavioral responses. The example plot in Figure 2 is helpful. I would like to see such plots for other participants, especially since the distribution of behavioral response times in this patient seem to be uncharacteristic of the overall times shown in the supplemental material.'*

We agree with the reviewer that seeing more examples will make the reader more confident that oscillations are present in the behavioral data, and in fact visible with the bare eye in many cases. We have therefore added examples to the Supplemental Information (the new Figure S2), showing two different participants for each task phase.

Please note that the example shown in Figure 2 is a representative example of the response distributions for the Catch-with-retrieval phase: the 90% central responses for this phase fell between 0.97 and 4.79 s after retrieval cue onset, with an average response at 2.24 s across on average 217.66 button presses from all 155 participants. The example participant in Figure 2 had 197 responses, with an average response time of 2.71 s, and a 90% range of 1.20 - 5.36 s.

1C. *'Another step that might help overcome skepticism is to fit the resulting oscillation from the FFT extraction to the behavioral histogram and show that the periodicity lines up with the behavior (perhaps with a goodness of fit test). I understand the analysis in Figure 4 addresses this point to some degree, although the degrees of freedom are quite high for the behavioral effect due to combining all trials into a single test (I think).'*

We have added a fit of a sine wave to the example shown in Figure 2A and the new examples in Figure S2 (light blue lines): $y(t) = \sin(2\pi ft + \theta)$. This sine fit used the frequency identified by the O-score method, but the phase offset θ was fit to the detrended response density trace (dashed green line). These fits show high overlap with the detrended response density traces for examples where the O-scores were high. We did not perform any further statistical analyses on this fit, because of the following reasons: 1) this would lead to circularity: the goodness-of-fit of the frequency would be determined based on the same data that were used to identify the frequency and its significance in the first place; and 2) the fit quality relies not only on an appropriate detection of the frequency by the O-score, but also on the stationarity in the response signal: the more stationary (effectively: the more spread out the responses of the participants), the better the fit quality is expected to be. The behavioral response signals are not stationary, as they have a clear onset and a (tapered) offset, but this varies between participants, making it hard to interpret any goodness-of-fit comparisons in this context. The O-score method takes care of this problem by using the auto-correlation histogram (see next answer). We do agree that providing the fit itself can instill further confidence in the O-score method.

We agree with the reviewer that Figure 4 provides a further illustration of the locking of the behavioral responses, although the main aim of this figure is to explore the differences between correct and incorrect trials. The pooling of the data indeed leads to a relatively high number of degrees of freedom for the individual V-test analyses, and a warning indicating this is given on lines 271-272. However, this potential inflation is corrected for in the subsequent analyses that compare the correct and incorrect trials. Firstly, the permutation test described in lines 281-286 generates shuffled distributions which contains the same number of degrees of freedom as the observed datasets, and therefore demonstrates that the observed difference between correct and incorrect trials was not the result of the number of degrees of freedom. Secondly, the difference was also assessed in a subset of participants by downsampling the number of correct responses to equate the number of incorrect trials. This resulted in above-zero V-statistics for correct trials without data pooling (i.e., locking), but not for incorrect trials, with the V-statistics for correct trials significantly higher than for the incorrect trials (lines 287-294).

1D. *'The hardest part of this method to wrap my head around is the use of the autocorrelation result to fit the FFT. Applying autocorrelation to a spike train that is essentially a continuous time series makes sense, but the behavioral responses are discontinuous in time. The result in Fig 2A makes it look like you can run the FFT on the response histograms themselves, which would be more convincing.'*

Firstly, we hope to clarify our approach: We apply the autocorrelation on the behavioral response train, after which the FFT is taken of the minimally smoothed upper half of the autocorrelation histogram. In doing this, we follow the exact pipeline proposed by Muresan and colleagues for spike trains. It is true that unlike the spike trains, the button press trains are obtained from events that are separated in time. However, it is important to note that from a mathematical point of view, the (temporal) origin of the events in the signal, spikes or button presses, does not matter to the execution of the autocorrelation; the mathematical tool itself does not require any temporal dependence in the signal in order to be valid. We do not interpret the result from the autocorrelation procedure directly, instead we use it as an intermediate step in the computation of the O-score. We do however predict (and our PPC analyses support this) that a phase reset in the slow rhythm involved in memory processing generates a temporal dependency in the button press trains, even though they come from different trials.

The autocorrelation is an important step in the computation of the O-score for several reasons.

1. The FFT assumes stationarity of the signal it is applied to. Our raw 'button press trains' are unlikely to meet this assumption, making direct application of an FFT problematic. This problem is also likely to occur in spike trains, particularly when these are recorded in a stimulus-locked way. Using the autocorrelation histogram (ACH) instead circumvents this problem.
2. The button presses (as is often the case for spikes) occur sparsely in time, which biases the FFT to higher frequencies if applied directly to the raw data. This can to some extent be avoided by using smoothing with a wide kernel and a larger time step, but these methods can introduce biases themselves and require careful testing of the kernel/step width to reach a balance between bias and sensitivity. Taking the FFT of the ACH reduces the bias instead by using a more densely packed signal.
3. Relating to the previous point, the ACH emphasizes patterns in the data that re-occur in time. This 'pulls out' oscillations compared to noise, making the method more sensitive to

oscillatory patterns. But perhaps more importantly, it avoids over-interpretation when oscillation-like events occur for only one or a few cycles.

Applying the FFT to the ACH, instead of the raw data, therefore is expected to make the method more sensitive AND more selective to oscillatory patterns. To illustrate this, we repeated some of the simulations now presented in Figure S10C of the paper. We used the simulations, because both the strength and frequency of the oscillations is known and can therefore be used to validate the detection method. Instead of applying the FFT to the upper half of the smoothed ACH (after removing the central peak), we applied the FFT directly to the raw data trace, after slight smoothing (the same as used for the ACH: Gaussian kernel with a standard deviation of 2 ms). The results are shown below, with the original method (from Figure S10C) on the left and the method without ACH on the right. As can be seen from the O-scores (top row), the **selectivity** of the method was severely reduced, leading to (marginally) significant O-scores when oscillations were weak or absent (upper right panel, indicated by asterisks on the left side of the x-axis, where amplitude is low and no oscillation is thus present in the simulated signal). The frequency plot (second row) shows that this was due to the detection of higher frequencies. At the same time, for strong oscillations (on the right side of the x-axis), the **sensitivity** of the method was severely reduced, leading to low O-scores even when all events followed the imposed periodicity. Both types of errors – false positives and false negatives – are highly undesirable. This can probably be improved by making other choices for the parameters (most notably, for the selectivity aspect, the number of permutations used to determine significance), but would most likely require a substantially different processing pipeline for the behavioral data. In conclusion, using the ACH as an intermediate step in the O-score method prevented the method from detecting spurious oscillations and made it more sensitive to the amplitude of stronger oscillations.

1E. *'The control analysis applied to visual recognition has a very different distribution of response times that are smaller and more highly constrained, so I am not sure it is an effective control.'*

We shared the reviewer's concern that the differences in response distributions between the task phases could have a potential impact on the O-scores. We therefore used two ways to check for any such impact. Firstly, we include time series length (i.e., the difference between the last included response and the first included response, for each participant) as regressor in the linear mixed-effects models used to compare the O-scores of the task phases with each other (see lines 201-205). Secondly, we performed an additional analysis for the Visual and Catch-after-retrieval task phases (which had the most compact response distributions). This control analysis loosened the frequency boundaries set by the O-score method, as described in the main text on lines 238-244, and the results are shown in Figure S5. We found similar O-scores and frequency distributions for the control analysis as for our original analyses. Finally, we conducted simulations which demonstrated that this additional control should reveal oscillations if they were present at similar strengths in the Visual task as in the Retrieval task (see Figure S10). The clear absence of significant O-scores for this control analysis for the Visual task therefore cannot be caused by a difference in RT distribution alone.

1F. *'Additional detail regarding how the shuffle distributions were constructed for the correct trials would also be helpful.'*

We have checked the relevant sections of the Results and Methods sections for completeness and have extended them where necessary. The relevant sections now read:

Results: *'... we fitted a trend curve (Gamma distribution) for each participant and generated 500 random series of button presses based on this structure, with the same number of data points as the original dataset (Figure 2B, II, see Methods for details).'* (lines 160-162)

Methods: *'...we randomly generated 500 time series for each participant matching the trial count and overall response density function of the participant's original button presses. First, a gamma probability function $r_{\text{gamma}}(t)$ was fitted to the participant's response distribution using the MATLAB built-in fitdist function (for an example see the grey lines in Figure 2), and scaled to the number of responses of the participant. We then generated 500 Poisson time series, with the probability of a response in a time step $\Delta t = 0.5$ ms, given by:*

$$P_{\text{resp}}(t \rightarrow t + \Delta t) = r_{\text{gamma}}(t) \Delta t.$$

If a gamma distribution could not be fitted (as assessed through a χ^2 goodness-of-fit test with $\alpha = 0.05$), the participant's button presses were instead randomly redistributed in time, with the new time per button press uniformly drawn from a window defined by one period of the participant's peak frequency and centered around the time of the original button press. Redistributing within one period of the identified oscillation ensured that this oscillation frequency of interest was not maintained in the reference data set, while minimizing changes to the overall response distribution.' (lines 951-964)

1G. *'The second half of the paper looks at LFPs from BF microwires, showing a phase reset effect. This result has been demonstrated during associative memory and other paradigms, including a success effect. These results add additional support to these previous findings, and certainly phase reset is a necessary condition for their behavioral analysis to work. While these findings are interesting, they occur in a relatively small group of patients - does the success effect for phase result reflect random effects analysis?'*

Yes, in principle our analysis approach includes random effects. As a clarification: the statistics we performed for Figure 5 were done at the level of microwire bundle, instead of patient, because analysis of the ERPs showed that the bundle level introduced as much variance as the patient level (i.e., while the microwires within a bundle had high dependency, bundles within a patient were as 'independent' as bundles from two different patients). We have now added this analysis as Supplementary Figure S8A. The statistical analyses we performed on the PPC data were based on the following: a paired samples t-test across bundles, which essentially identifies the fixed effect for correct/incorrect trials or the contrast between them, ignoring any bundle level differences in the mean response. These paired samples t-tests were entered into a permutation-based analysis to identify statistically significant clusters in the time-frequency dimensions, i.e. the results of the t-tests were compared with shuffled data from the same bundles (either time-shuffled or trial-shuffled). The permutation-based approach does not require any assumptions about the nature of the effect or the distributions of these effects. More importantly, the reference distributions obtained through permutations maintains any intra- and inter-subject (or -bundle) variability present in the dataset that is not of interest, and hence avoids inferences based on these sources of variability. Our permutation-based approach is therefore much less likely to be driven by chance differences within one or a few patients than other statistical methods that require assumptions about the reference distribution. In addition, we Z-scored the PPC data against the pre-trial baseline, to further avoid inferences based on strong inter-subject or inter-bundle differences. We therefore think it is unlikely that our PPC results are obtained by chance or by outliers. This is strengthened by two observations: 1) visual inspection of the PPC results of individual patients (Figure S6 E&F) show the same overall patterns as the average: higher PPC for correct than for incorrect trials; and 2) we performed a new PPC analysis of the data from the macro contacts of these patients in response to the comments of reviewer #2 (please see response to point 2A and Figure S9). These analyses were performed with patients as the second level, as inter-bundle variability for these data was higher than inter-subject variability (see Figure S8B). This patient-level analysis produced qualitatively similar results.

1H. *'Do these same 10 subjects exhibit periodicity in their behavioral responses that matches the frequency of maximal phase reset? That would be especially interesting.'*

We would like to refer the reviewer to Figure S7, which contains a direct comparison of the O-score and PPC analyses for the 10 patients. Overall, there appears to be a similarity between the peak frequencies found by both methods across the three task phases available for these patients. However, we urge extreme caution in interpreting the correlations between these measures, as they each only contain 10 data points. For this reason, we have chosen to refrain from making claims about this relationship in the paper. More conclusive statements will require larger iEEG datasets with more patients and/or intervention techniques that allow for altering the hippocampal rhythm, which we discuss in lines 482-501 of the Discussion section.

Reviewer #2:

'This manuscript investigates theta oscillations and their behavioral relevance in human memory tasks using a clever behavioral analysis along with intracerebral recordings. The authors analyzed button responses during both memory and visual perception tasks using the O score, a method adapted from spike train analysis, and found evidence for slow-theta rhythmicity in these responses only during memory tasks. Correct memory was associated with hippocampal phase consistency in this slow theta band for correct trials. Overall, this manuscript is clearly written, well-powered, and uses appropriate methods to test novel predictions about theta oscillations. I did not find too many weaknesses and believe this manuscript will be impactful in the field. Nonetheless, I provide below several points which could improve the manuscript and make it suitable for publication.'

Major comments

2A. *'The authors report neural results from micro LFP contacts and this is well justified in my view. On the other hand, the majority of human iEEG studies have analyzed data from macro contacts. It would therefore be helpful to readers to include results from the macro contacts as a supplemental analysis. Do results differ between macro and micro-contacts?'*

In response to this comment, we pre-processed the hippocampal macro contacts from the same patients, performed the PPC analyses on these data, and included these results in the supplementary information (Section 3 and 4, Figure S9). Unlike for the microwires, we performed the second level statistics for the macro contacts at the subject level, because a control analysis suggested that patient is the major source of variance for the macro data (new Figure S8B; this analysis is also explained in response 1G). This is likely to reduce the power of the analyses. Despite this difference, the PPC results of the macro contacts clearly show qualitatively similar increases in PPC as the microwires, dominant in the same frequency range and extending over much of the memory-dependent trial periods.

2B. *'Human theta can show substantial variability between subjects and channels. Was there relationship between the frequency of O score and frequency of peak PPC across subjects?'*

We did directly compare the peak frequencies from the O-score and PPC from these patients, please see Figure S7. In general, the peak frequencies from both methods seem to follow similar patterns, and this was the case across the three task phases that these patients completed. We have chosen not to make any conclusive statements about or provide statistics for this analysis in the manuscript, because each of these correlations only contain 10 data points (patients), which is too few to interpret these correlations with confidence. Please also see our response 1H to a comment from reviewer #1.

2C. *'Similarly, did the authors observe any anterior-posterior or laterality differences in PPC?'*

We have added the PPC results for correct trials split into anterior, middle and posterior hippocampus in Figure S6H, and left and right hippocampus in Figure S6I. There are trends visible in the anterior-posterior split, namely a focus on encoding in the anterior hippocampus and a focus on retrieval in the posterior hippocampus, but we think our dataset does not have enough power (i.e., a sufficiently large number of bundles/contacts in each sub-region) to analyse these differences in a statistically meaningful way in further detail. There are no apparent differences between the hemispheres. We have added references to these new SI figures in the main text on lines 347-349.

2D. *'The authors found no effects in visual tasks. On one hand, this is a nice demonstration of the memory-dependence of their effects, but another line of literature has shown modulation of hippocampal LFPs with eye movements (e.g. Jutras et al., Kragel et al., Hoffman et al.). I suggest the authors include some description of the key task differences between studies in the discussion.'*

We have added a sentence clarifying the differences between the memory-independent task phases used here and the tasks used to show links between eye movements and hippocampal theta rhythms in the Discussion. We also clarify the differences with previous studies demonstrating theta-rhythmicity in behaviour during visual attention tasks. The relevant section of the Discussion reads:

'We did not observe significant oscillations in behavior for processes that we a priori marked as memory-independent, namely answering the catch questions after reinstatement and the visual task. These task phases also did not contain an attentional selection element and did not rely on memory-guided visual search, which are cognitive processes that have previously been linked to theta rhythmic modulation of behavior³⁷⁻⁴² and saccadic eye movements⁴⁸⁻⁵⁰, respectively.' (lines 446-450)

2E. *'I am concerned by the high channel count (n=326 channels, Figure 6 caption; Table S6) from a small number of patients. It is doubtful that the signals on channels are independent and therefore the degrees of freedom are likely over-inflated.'*

We agree with the reviewer that it is unlikely that microwires from the same bundle are independent. Prior to performing the PPC analyses in Figure 5, we formally analysed the correlations between ERPs from different wires and compared this to correlations between bundle averages within and between patients. This indeed showed high correlations between wires of the same bundle, but not between bundles from the same patient. We have now added this analysis to the Supplementary Information (Figure S8A). This analysis was the reason why we performed the statistics for the PPC analyses on the averaged PPC maps across the microwires in a bundle, hence avoiding this problem in Figure 5. We apologise if this did not become clear from our previous descriptions of the methods. We opted for a different approach for the phase analyses in Figure 6, because even though channels can be highly dependent within a bundle, there can be phase flips between neighbouring channels on the same bundle, depending on their implantation depth in hippocampus. It is not possible to identify the exact location of the microwires on MRI scans, so phase flips can only be identified based on the data itself, which would lead to circularity in the analysis. We therefore used a permutation test to validate our phase findings, which uses the same

number of data points and therefore corrects for inflation of the degrees of freedom. This indeed resulted in a change in interpretation for the stimulus-locked analysis. To avoid over-interpretation, we have removed the results from the original V-tests (with the uncorrected high datapoint count), from Figure 6, leaving only the results from the permutation test. We have also made the possibility of inflated degrees of freedom explicit in the main text.

The relevant text now reads:

‘Both analyses provided support for a half-cycle difference between encoding and retrieval (V-test around 180 degrees; cue/stimulus-locked: $V=33.1$; $p=0.0047$; response-locked: $V=46.7$; $p=0.0013$; $n=326$). We tested whether the observed non-uniformity around 180 degrees phase difference, which could be inflated due to the high channel count, was expected by comparing the V-statistics to those from 500 time-shuffled datasets. We conclude that phase opposition for the response-locked trials was unlikely to be obtained by chance ($p=0.042$), while for stimulus/cue-locked data ($p=0.13$), the observed V-statistic could, in part, be inflated by channel count or a phase bias, for example due to asymmetry in the theta cycles⁴⁷.’ (lines 388-395)

For clarity, we have also added the number of datapoints to Figure 5 and the new SI Figures S6H and S6I.

Minor comments

2F. *‘Is zero radians the peak or trough in the figures? Please add these labels, as different groups use different conventions (unfortunately!).’*

We thank the reviewer for pointing this out. We have added the definition to the caption of Figure 4. Please note that Figure 6A and B show phase differences, therefore this definition is not applicable there.

2G. *‘Did the controls and patients differ on O score? Could these methods be used to identify MTL dysfunction in epilepsy patients in lieu of intracranial monitoring?’*

This is an interesting question, but our data do not suggest O-scores are different between epilepsy patients and healthy volunteers, see Figure S4E (compare experiments 12-13 with all other experiments), in addition to a relatively similar task performance. To formally address this question, we fitted a linear-mixed model to the Z-scored O-score data from participants in memory group 1 (which includes the patients and all healthy participants who performed the same memory task version), with epilepsy diagnosis as fixed effect and intercept per subject as random effect. This revealed no difference between the O-scores of epilepsy patients and the healthy volunteers (coefficient=-0.074; 95% CI: -0.22-0.077; $t(319)=-0.96$; $p=0.34$). Adding the epilepsy diagnosis as fixed effect to the full model (described in the main text on lines 202-206) also showed no effect on O-score (coefficient=-0.062; 95% CI: -0.27-0.14; $t(555)=-0.59$; $p=0.55$), and did not affect the outcomes for the memory dependency (coefficient=0.27; 95% CI: 0.19-0.36; $t(555)=6.30$; $p<0.0001$) and reaction time dependency (coefficient=0.0035; 95% CI: -0.0036-0.011; $t(555)=0.97$; $p=0.33$). O-scores thus strongly overlap between patients and controls, with no indication they would be useful as a diagnostic tool.

REVIEWER COMMENTS

Reviewer #1 (Remarks to the Author):

The authors have performed a series of revisions to the description of the O score modeling, including expanded simulation data, that have certainly improved the manuscript. I think the core behavioral result remains highly novel and interesting. Adding the sine fits for the predicted oscillation I think helps explicate the findings.

I have one remaining questions re: the O score modeling. The frequency bounds differ significantly from Muresan et al, who show that O scores for low oscillations are higher than for higher frequencies because of the higher magnitudes of low frequency peaks relative to the baseline spectrum. This issue would influence the specific frequency at which the authors find the O score identifies an oscillation, but would not affect the comparisons between conditions (eg recall success vs failure). The main weakness of the paper is that the authors are not able to apply the O score calculation to incorrect trials (low trial number), although they attempt to overcome this via fitting the oscillation from the correct trials to the incorrect trials. They are fairly explicit in the Results section about this limitation, but the abstract is somewhat misleading and I think mention of the nonrecalled difference should be removed from there (esp since only a limited number of participants contribute the non rec data). For Figure 4, the authors should show downsampled response time distributions from the downsampled recalled trials, so readers can be confident of this step (rather than just the V statistics).

The other weakness then is that the differences for memory-related vs visual trials (the latter of which did not show the same O score results in terms of oscillations) relies on a comparison to a separate group of subjects who did not perform the memory task (the catch-after-retrieval distribution doesn't really look different than the encoding/retrieval distributions in terms of O score results, so the claim regarding memory-specific oscillation for O score relies on this comparison to the visual data). The distinct subject groups is an important point to clearly state in the Results when presenting the O score findings I think.

I have one additional question re: the O score fitting. To generate a reference O-score distribution the authors use two different methods without specifying why certain subjects had gamma distributions that could not be fitted. Why choose to use two separate methods rather than randomly redistribute each participant's button presses in time (the second method, if that one works)? Can the authors show that the latter method that redistributes button press times does not significantly change the reference distributions for subjects who met the defined goodness-of-fit criteria for the gamma probability function model?

Finally, while I understand why the authors include the phase reset data from 10 subjects with microwires, but given the limitations in drawing any conclusions from these data (which the authors acknowledge), I think the main findings of the paper would be better served by removing the electrophysiological data, as the presence of memory-relevant phase reset in the hippocampus is readily demonstrated in the existing literature.

Reviewer #2 (Remarks to the Author):

The authors have done a nice job revising their manuscript and have addressed my concerns to my satisfaction. I now believe this manuscript should proceed to publication.

Response to reviewers' comments

Reviewer #1:

'The authors have performed a series of revisions to the description of the O score modeling, including expanded simulation data, that have certainly improved the manuscript. I think the core behavioral result remains highly novel and interesting. Adding the sine fits for the predicted oscillation I think helps explicate the findings.'

1A. 'I have one remaining questions re: the O score modeling. The frequency bounds differ significantly from Muresan et al, who show that O scores for low oscillations are higher than for higher frequencies because of the higher magnitudes of low frequency peaks relative to the baseline spectrum. This issue would influence the specific frequency at which the authors find the O score identifies an oscillation, but would not affect the comparisons between conditions (eg recall success vs failure).'

The reviewer is correct in stating that we use a different frequency band of interest than the example given in the paper by Muresan et al.. However, as they state in their paper, this frequency band of interest is a choice that is to be made by the experimenter, depending on the frequency band of interest that is central to their hypothesis. Given that we expected any behavioral oscillation to occur at relatively low frequencies, but had no strong hypotheses about the precise frequency band, we *a priori* set our frequency band of interest to between 0.5 and 40 Hz. Following the procedure described by Muresan et al., and explained on lines 927-933 of the Methods section, this

frequency band was tested for feasibility against the properties of the data, and automatically adjusted where necessary, for every participant individually. This ensured we did not identify frequencies above the Nyquist frequency, or below the minimum required number of 3 cycles, as defined by Muresan et al.. We have now clarified the wording around the definition of the frequency band of interest in the Methods section (lines 924-927):

'The O-score analysis requires the experimenter to define a frequency range of interest. We *a priori* defined a wide frequency range of interest of between $\min_{\text{init}}=0.5$ and $\max_{\text{init}}=40$ Hz, as we did not want to limit the analyses to a specific frequency band, yet did not expect to detect frequencies in the higher gamma frequency band.'

It is also true that Muresan and colleagues reported higher raw O-score values for lower oscillation frequencies for their spiking data. As they argue on page 1341, this is likely due to physiological principles: slower oscillations engage a larger part of neural tissue, recruit more spikes and hence have higher amplitudes, which is reflected in the higher O-scores. However, we could not be sure this applied to our behavioral data, and we carefully considered potential frequency biases while designing our analyses. In the reaction time data, frequency biases can potentially occur simply due to the limited number of reaction times and the typical concentration of these responses around 1 or 2 seconds after cue or stimulus onset (see the green response density curves in Figure 2 and S2). To mitigate any biases resulting from this concentration of responses (and any differences in O-scores between frequencies resulting from it) we introduced the participant-specific and frequency-specific Z-scoring procedure, explained in Figure 2B and lines 157-166 and 949-978. This Z-scoring procedure is based on participant-specific reference distributions which, due to the gamma fit (please also see answer 1E), use a closely matching overall response density trend, as well as the exact same number of responses as the intact dataset. We analysed the O-scores for each of the 500 randomly generated response traces *at the same frequency* as identified for the intact data, ensuring that any frequency bias due to the response trend and number of datapoints was accounted for in the reference O-scores. A high O-score resulting from any of these factors would therefore receive a low Z-score, because the bias would also be present in the reference distribution, which would therefore also contain high O-scores. We tested this procedure using our simulations (Figure S12) for different ground-truth frequencies between 2.5 and 15 Hz and found identical performance of the Z-scored O-scores for different frequencies, demonstrating that bias did not affect the comparison of Z-scored O-scores between different frequencies.

1B. The main weakness of the paper is that the authors are not able to apply the O score calculation to incorrect trials (low trial number), although they attempt to overcome this via fitting the oscillation from the correct trials to the incorrect trials. They are fairly explicit in the Results section about this limitation, but the abstract is somewhat misleading and I think mention of the nonrecalled difference should be removed from there (esp since only a limited number of participants contribute the non rec data).

We have removed the reference to the incorrect trials from the abstract (please see also our response to point 1C).

1C. For Figure 4, the authors should show downsampled response time distributions from the downsampled recalled trials, so readers can be confident of this step (rather than just the V statistics).

We have now added the phase distributions for the down-sampled datasets to the Supplementary Information in Figure S7. These figures show the mean phase of response distribution for down-sampled correct and incorrect trials and the variability across participants who contributed to the data and statistics in the right panels of Figure 4. As can be appreciated when comparing Figure 4 with Figure S7, the differences between the two are minimal, and both statistical approaches (i.e., shuffling correct and incorrect labels and down-sampling within participants) support the conclusion that incorrect trials do not lock to the oscillation of the correct trials.

1D. The other weakness then is that the differences for memory-related vs visual trials (the latter of which did not show the same O score results in terms of oscillations) relies on a comparison to a separate group of subjects who did not perform the memory task (the catch-after-retrieval distribution doesn't really look different than the encoding/retrieval distributions in terms of O score results, so the claim regarding memory-specific oscillation for O score relies on this comparison to the visual data). The distinct subject groups is an important point to clearly state in the Results when presenting the O score findings I think.

We have added a sentence stressing this to the Results section on lines 196-198:

'Note that the Catch-after-retrieval data were obtained from the participants in memory task group 1, while the Visual task was recorded in an independent group of participants (see Figure 1C).'

Please note that it is not possible to perform the memory task and visual task in the same participant, as each task relies on the participants being naïve to the presented material. A between-subject design was therefore the only feasible approach. As with any between-subject design, this comes with larger variability, and typically larger samples sizes are required to show between-group effects. Given the large size of both the memory and visual datasets and the well-matched demographics, it is extremely unlikely that differences between participants drove the reported O-score differences. The fact that we replicated the difference between memory-dependent task phases and the visual task with 4 different stimulus sets, supports this conviction further.

1E. I have one additional question re: the O score fitting. To generate a reference O-score distribution the authors use two different methods without specifying why certain subjects had gamma distributions that could not be fitted. Why choose to use two separate methods rather than randomly redistribute each participant's button presses in time (the second method, if that one works)? Can the authors show that the latter method that redistributes button press times does not significantly change the reference distributions for subjects who met the defined goodness-of-fit criteria for the gamma probability function model?

As we explained in the answer to comment 1A, we felt that it could be important to correct for the overall response distribution of the reaction times; we expected the reaction time distributions to be non-uniform (and probably slightly less uniform than spike trains). Because of the overall response distribution, we expected the spectrum of the reaction times to be non-white, even in the absence of an oscillation, which could potentially introduce a frequency bias. As previously explained, we used the Z-scoring procedure to combat any such biases. This procedure compares the O-score from the original data against O-scores from reference datasets with the same characteristics and at the

same frequency. For such a procedure to work well, the reference datasets need to be as similar as possible to the original data in all aspects except for the characteristic of interest, which here is the presence of an oscillation. We therefore aimed to generate reference distributions based on the same response density structure as the original data, for each participant individually, by fitting this distribution and generating reference dataset based on the fit.

The gamma distribution fit had a high overall success rate across task phases and participants; we have now added a characterization of the fraction of successful fits to the Supplementary Information in Figure S5A. As expected, where the fit failed, this was most likely caused by a low density of responses (i.e., few button presses per second), see Figure S5B, which most often occurred in the Encoding task phase, due to the long response times and low number of button presses per participant for this task phase (see also Table S3).

We agree with the reviewer that any confounding effects due to participants whose response distribution could not be fitted should be avoided. We therefore re-computed the statistics underlying the results in Figure 3A while excluding participants for whom the gamma fit was unsuccessful. The results are shown in the new Figure S5C and corresponding statistics are shown in the figure and the caption. Excluding participants for whom the gamma fit failed produced highly similar results and did not affect the conclusions drawn.

We also considered the reviewer’s suggestion to only use uniformly shuffled reference datasets. We expect such reference distributions to be less successful at removing any frequency-related O-score biases, as unlike the original dataset, this reference distribution will have relative uniform frequency spectra. This is therefore expected to inflate the Z-scores of the O-scores of individual participants, and in turn, the statistics across participants. Indeed, when using pure shuffling to generate reference distributions for all participants, second level T-scores increased for all task phases as shown in the figures below. As this approach led to a predictable inflation of the statistics, we have opted not to add it to the main manuscript, and maintain the more conservative approach using gamma fits.

A: Second level t-score for the original dataset (left, corresponding to Figure 3A) and for a dataset where the O-score reference distributions were generated without fitting a trend, i.e. with using shuffling only (right). Second level t-scores increases for all task phases when the trend fit was removed from the Z-scoring procedure; **B:** Differences between the second level t-scores shown in A,

i.e., the no trend fit data minus the original data. The differences were positive for all task phases, with the largest difference for the Catch-with-retrieval phase.

1F. Finally, while I understand why the authors include the phase reset data from 10 subjects with microwires, but given the limitations in drawing any conclusions from these data (which the authors acknowledge), I think the main findings of the paper would be better served by removing the electrophysiological data, as the presence of memory-relevant phase reset in the hippocampus is readily demonstrated in the existing literature.

Even though the intracranial EEG data we present have some limitations, mostly due to the rare nature of these data and clinical restrictions as in any iEEG study, we feel that the electrophysiological data add several important elements to both our manuscript and the wider literature. We have outlined them below:

- 1) We feel that it is important to demonstrate the extended phase consistency in the specific memory task that was also used to observe the behavioral oscillations. Inferring from existing work, using different task settings, that such phase locking would likely be present in our memory task would in our view provide a much weaker argument.
- 2) In addition to point (1), including the iEEG data allows us to compare successful and unsuccessful trials both in the electrophysiological signals and in behavior. Both comparisons show a difference in oscillatory strength in the lower theta frequency band, and we believe that this parallel effect in electrophysiology and behavior strengthens the point that theta phase locking is memory relevant.
- 3) While phase resets in memory paradigms have been reported before, we are aware of only two studies reporting a difference between correctly remembered and forgotten trials (Kota et al., 2020 and Fell et al., 2008). These studies both reported phase locking differences after cue/stimulus onset during encoding and retrieval, but neither looked at phase consistency differences relative to the moment in the trial where encoding/reinstatement actually took place. Our task design, with the subjective encoding and retrieval button presses, allowed us to demonstrate that phase locking extends in time up to the moment where a memory-informed decision takes place. This temporally extended phase consistency provides an important mechanistic underpinning of the behavioral findings, as it demonstrates that phase resets could in fact lead to phase locking around the time of encoding or retrieval button press;
- 4) Phase resets have, to our knowledge, never been demonstrated using microwires in human hippocampus. These microwires pick up extremely local signals, when compared to the much larger macro electrodes, making this finding relevant to the wider literature. With the addition of the macro data, added based on comments from reviewer #2, we are the first to show phase resets on both microwires and macro-contacts in the same patients.

Based on these reasons, we believe that the iEEG data are an important piece of the puzzle, and if possible, would like to maintain them in the main manuscript. We have however altered several sentences in the Results section to emphasize that the finding of a theta phase reset is not novel per se (see lines 322-327 and 362-368), but that we add to the literature by demonstrating that the phase consistency extends until the memory-dependent button press, and that this is an important physiological underpinning of the core behavioural results.

Reviewer #2:

'The authors have done a nice job revising their manuscript and have addressed my concerns to my satisfaction. I now believe this manuscript should proceed to publication.'

We thank the reviewer again for their thoughtful and constructive contributions.

REVIEWER COMMENTS

Reviewer #1 (Remarks to the Author):

I am satisfied with the authors' response to my questions. I give them credit for developing such a complex and nuanced analysis of behavioral data.